# Evidence for oxygen-conserving diamond formation in redox-buffered subducted oceanic crust sampled as eclogite

Sonja Aulbach [1✉] & Thomas Stachel[2]

Cratonic eclogite is the product of oceanic crust subduction into the subcontinental lithospheric mantle, and it also is a fertile diamond source rock. In contrast to matrix minerals in magma-borne xenoliths, inclusions in diamond are shielded from external fluids, retaining more pristine information on the state of the eclogite source at the time of encapsulation. Vanadium is a multi-valent element and a widely used elemental redox proxy. Here, we show that that xenolithic garnet has lower average V abundances than garnet inclusions. This partly reflects crystal-chemical controls, whereby higher average temperatures recorded by inclusions, accompanied by enhanced $Na_2O$ and $TiO_2$ partitioning into garnet, facilitate V incorporation at the expense of clinopyroxene. Unexpectedly, although diamond formation is strongly linked to metasomatism and xenoliths remained open systems, V concentrations are similar for bulk eclogites reconstructed from inclusions and from xenoliths. This suggests an oxygen-conserving mechanism for eclogitic diamond formation, and implies that eclogite is an efficient system to buffer $fO_2$ over aeons of lithospheric mantle modification by subduction-derived and other fluids.

[1] Institut für Geowissenschaften, Goethe-Universität, Frankfurt am Main, Germany. [2] Earth and Atmospheric Sciences, University of Alberta, Edmonton, AB, Canada. ✉email: s.aulbach@em.uni-frankfurt.de

Oxygen fugacity ($f$O$_2$) is an intensive variable that controls the speciation and mobility of carbon and therefore its cycling among Earth's reservoirs[1]; it is, therefore, often invoked to play a critical role in diamond formation[2–4]. Kimberlite-borne eclogite xenoliths have protoliths that formed during partial melting of dry peridotite, probably at Mesoarchaean to Palaeoproterozoic spreading ridges, and can therefore provide invaluable information on the oxygen fugacity ($f$O$_2$) of the convecting mantle and the evolution of this parameter in deep time[5]. Like modern mid-ocean ridge basalts (MORB), the ancient protoliths underwent seawater alteration and metamorphism in subduction zones[6], in part accompanied by diamond formation[7]. The compositions of isolated minerals entrapped in their chemically inert diamond host freeze-in concentrations that reflect temperatures and garnet-clinopyroxene equilibria at the time of diamond formation. In contrast, xenoliths continue to evolve via interaction with fluids and melt that cause mantle metasomatism[8], giving rise to differences in the chemical composition of occluded minerals and their xenolithic counterparts[9] (Fig. 1a). However, the diamond-forming process is itself already metasomatic in nature, and may have been accompanied by changes in chemical composition and possibly redox state[3,10].

Vanadium is a multi-valent element that becomes less compatible in mantle minerals with increasing $f$O$_2$[11,12], and is now widely applied as a redox proxy to unravel $f$O$_2$ attending the formation of mantle-derived melts and their residues[11,13–15]. The development of the V redox proxy is particularly useful where conventional oxybarometry is difficult to apply, as in the case of isolated inclusions in a diamond. However, the simple mineralogy of xenolithic eclogites, composed of subequal proportions of garnet and clinopyroxene with a limited number of accessory minerals[6], and their corresponding diamond source rocks at depth[16], belies a complex evolution that would preclude a simple interpretation of mineral V abundances. Crystal-chemical controls on V incorporation in eclogite minerals are likely to play a role. This is important because inclusions in diamond (DI) are mostly unpaired (i.e., garnet or clinopyroxene only in a given eclogitic stone). Furthermore, apart from the redox-dependence of V partitioning, like any other trace element, V abundances in eclogite depend on accumulation-differentiation processes within the oceanic crust, and mobility during subduction-related melt extraction and later melt metasomatism. While the quantification V concentrations in xenolith minerals by laser-ablation microprobe inductively-coupled plasma mass spectrometry (LAM-ICPMS) is routine, this destructive analytical technique is not often applied to inclusions in the diamond, where V$_2$O$_3$ concentrations determined by electron probe microanalyser (EPMA) are reported instead in some studies.

Cratonic eclogite xenoliths have compositions and element relationships indicating that they originated from various parts of subducted oceanic crust, and eclogitic DI is generally interpreted as derived from cratonic eclogite[16]. Geochronological studies utilising radiogenic isotope systematics of DI show that eclogitic diamond formation occurred predominantly in the Mesoarchaean and Palaeoproterozoic[17,18]. Due to their multi-stage evolution, obtaining reliable dates for eclogite xenoliths is challenging, but nevertheless often successful (for a recent review of ages also for eclogite suites used in this study see ref. [19]). Xenoliths and DI show similar ages in various cratons, which in turn correlate with the known timing of Archaean craton amalgamation (Kaapvaal) or Palaeoproterozoic collisional processes at the margins of Archaean cratons (Slave). This has been taken to indicate that diamond formation in eclogite is temporally and genetically linked to the subduction and closure of ocean basins[7,14,17,20]. In contrast to most DI, the cratonic eclogite reservoir continued to evolve subsequent to diamond formation, mostly through

interaction with fluids and kimberlite-like small-volume melts[9]. This type of metasomatism affected 20–40% of the eclogite reservoir and caused diamond destruction at most xenolith localities[21], although it can occasionally be associated with abundant diamond growth (e.g., northern Slave craton[22]).

Although diamond formation is episodic[16], and younger overgrowths on the eclogitic diamond have been detected by cathodoluminescence and direct dating of diamonds in distinct zones[23,24], it is unlikely that the DI used in this study exclusively belong to a late generation of diamond, as also attested by the generally old ages derived from eclogitic inclusions[7,14]. Thus, it seems plausible that DI record a stage of evolution at least as old as, and likely predating that of, equivalent minerals in eclogite xenoliths, which additionally experienced subsequent mantle metasomatic events.

Here, we exploit published information to assess similarities and differences of V abundances in garnet and clinopyroxene included and protected in diamond, and their xenolithic counterparts, which remained open systems. Underpinned by quantitative modelling, these findings not only have implications for the $f$O$_2$ evolution of the subcontinental lithospheric mantle sources of the diamond but also provide intriguing insights into the temperature-, composition- and redox-dependent partitioning behaviour of V in metabasaltic systems, the understanding of which is fundamental to applying V in eclogitic minerals as a redox proxy.

## Results

**Literature data used to illustrate multi-stage eclogite petrogenesis.** Before V in isolated minerals or in bulk rocks can be confidently used as a redox indicator, the underlying controls on V abundances must be understood. To shed more light on this issue, xenoliths and DI are matched for six occurrences in four cratons (Supplementary Data 1). This accounts for the fact that different localities have endemic compositional systematics that reflects the individual origin and evolution of each eclogite reservoir. See Methods for detailed information on the database, an assessment of the two main analytical methods used for the determination of V concentration and the rationale for the reconstruction of bulk eclogite compositions.

Minerals and reconstructed bulk eclogites for xenoliths and corresponding DI show significant differences with respect to multiple variables, summarised in Table 1 and visualised as box-and-whiskers plots in Fig. 1. Various correlations, the suggested causative relationships of which are reviewed below, are summarised in Table 2 and visualised in Supplementary Figs. 1-4. The large scatter and weak resultant correlations reflect the multitude of origins and petrologic processes in eclogite petrogenesis, including, but not limited to (1) an origin as cumulates vs. complementary melts in palaeo-spreading ridges, (2) various degrees of differentiation and melt-rock interactions during the low-pressure stage, (3) various degrees of melt extraction upon subduction in warm Archaean subduction zones. In addition, xenoliths might have been affected by mantle metasomatism. Despite this complexity, we suggest that crystal-chemical, oxygen fugacity and petrologic controls are recognisable in the fact that statistically significant correlations, albeit weak, can still be isolated.

**Temperature and crystal-chemical effects on V distribution.** A striking observation of the dataset is that median V concentrations in garnet from xenoliths are significantly lower (110 ppm) than those in DI garnet (220 ppm) (Fig. 1b). The average valence and therefore compatibility of V depends on $f$O$_2$, and V becomes more incompatible in common mantle minerals

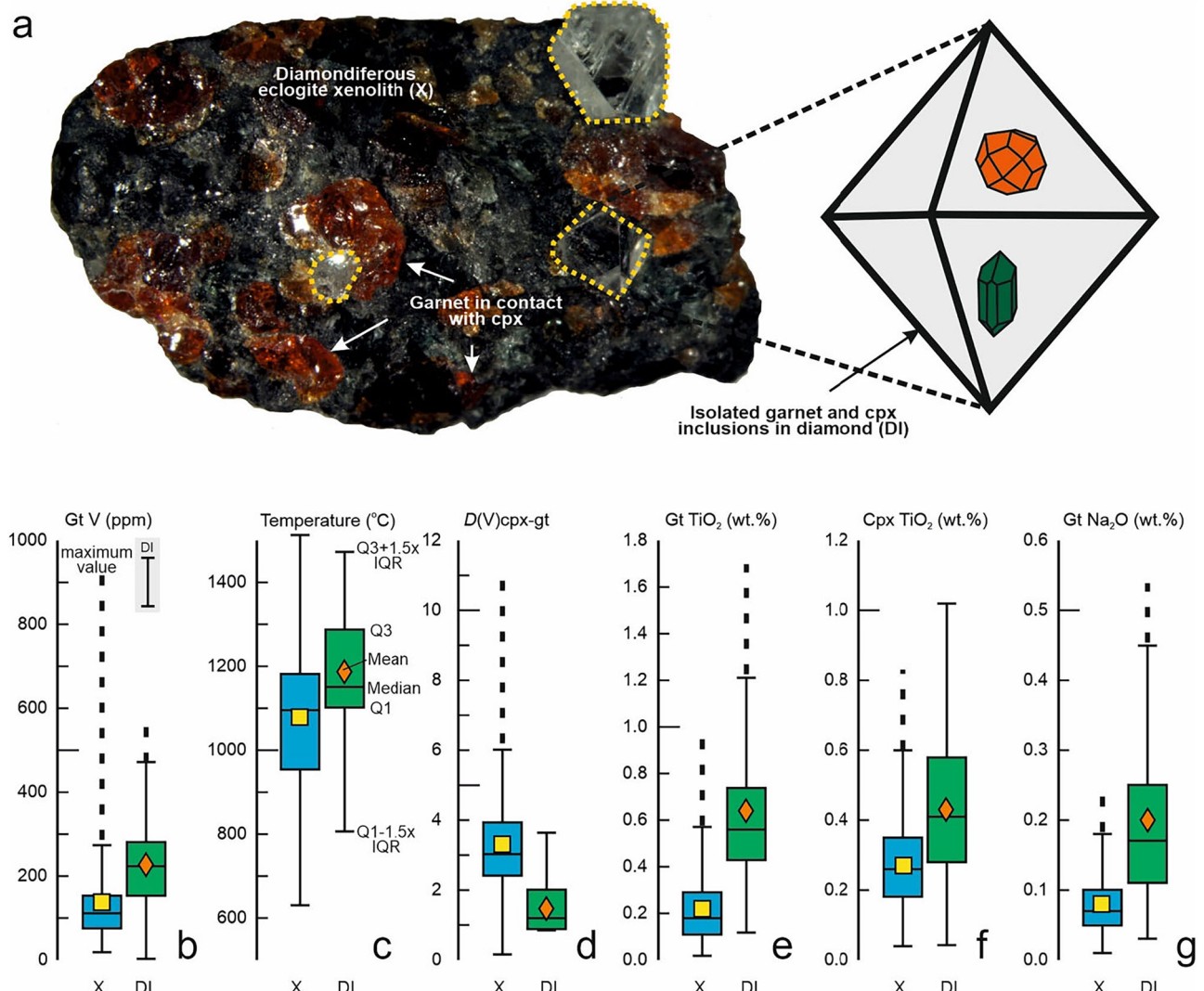

**Fig. 1 Illustration of the closed-system behaviour of isolated eclogitic inclusions in diamond (DI) vs. open-system behaviour of touching inclusions in eclogite xenoliths (X). a** Photograph of a diamondiferous eclogite xenolith from Fort à la Corne (~1.5 cm longest dimension); diamond crystals are highlighted in yellow stipples. Also shown is the schematic of a diamond hosting eclogitic garnet and clinopyroxene inclusions that are non-touching, which prevents them from chemically re-equilibrating to lower temperatures, while the inert diamond also protects them from communicating with the open-system matrix. Contrasting temperature estimates and compositions shown as box-and-whisker plots: **b** Vanadium concentrations (ppm) in garnet (gt), **c** temperature estimates (°C) reflecting those of the eclogite sources at the time of xenolith entrainment in kimberlite, vs. the time of encapsulation of non-touching inclusions in diamond, **d** Distribution (*D*) of V between clinopyroxene (cpx) and garnet ($D(V)_{cpx-gt}$), **e** $TiO_2$ content (wt.%) in garnet, **f** $TiO_2$ content (wt.%) in clinopyroxene, **g** $Na_2O$ content (wt.%) in garnet. Explanations for the statistical parameters displayed in this figure (mean, median, Q1 first quartile, Q3 third quartile) are shown in panel **c**. The whiskers are here defined as $Q3 + 1.5 \times IQR$ (interquartile range), and $Q1-1.5 \times IQR$ or the minimum value, whichever is higher; stippled lines extend to the maximum value if $>Q3 + 1.5 \times IQR$. Average 1σ uncertainty on V abundances in garnet from DI of 117 ppm is shown as error bar; average 1σ uncertainty on V abundances in garnet from xenoliths of 5.6 ppm is not displayed at the scale; both correspond to typical uncertainties for multiple analyses per sample reported in the literature, see Methods). Data sources in Supplementary Data 1.

with increasing $fO_2$[12]. As minerals in xenoliths may have interacted with external metasomatic melts, whereas DI are protected inside the inert diamond host, the lower concentration of V in xenolithic garnet relative to DI garnet could, at first glance, be misinterpreted as reflecting a reduction of V abundances due to oxidative metasomatism. However, a notable difference between garnet in eclogitic DI and xenoliths is the higher equilibration temperatures recorded by the former at the time of encapsulation in diamond (median of 1154 °C for DI vs. 1096 °C for xenoliths) (Fig. 1c) under generally warmer mantle conditions in Archaean and Palaeoproterozoic time. Indeed, in three of six sample suites, the distribution coefficient *D* between clinopyroxene and garnet ($D(V)_{cpx-gt}$) is significantly negatively

correlated with temperature (Supplementary Fig. 1a-b; Table 2), with higher median values for xenoliths (3.0) than for DI (1.2) (Fig. 1d). Despite the considerable scatter, this suggests an underlying temperature control on V incorporation in both clinopyroxene and garnet.

Rutile solubility in garnet from eclogite xenoliths shows a strong positive temperature dependence, such that garnet in high-temperature eclogites has higher $TiO_2$ contents[25], and rutile is known to be an important V host[26]. Median $TiO_2$ contents in DI garnet (0.56 wt.%) are higher than those in xenolith garnet (0.18 wt.%) (Fig. 1e), and $TiO_2$ in garnet is significantly positively correlated with temperature in all six sample suites (Table 2), similar to V abundances in garnet

**Table 1 Salient parameters of clinopyroxene, garnet and reconstructed whole rocks in eclogite xenoliths and inclusions in diamond (DI), as well as temperature-$fO_2$ estimates from the literature (references in Supplementary Data 1).**

| | Clinopyroxene | | | | | Cpx/Gt |
| --- | --- | --- | --- | --- | --- | --- |
| | $TiO_2$ | $Al_2O_3/FeO^t$ | Mg# | Jd | V | D(V) |
| Unit | wt.% | | | | ppm | |
| Xenoliths $n =$ | 261 | 261 | 261 | 261 | 261 | 261 |
| Min | 0.04 | 0.4 | 0.6 | 0.13 | 14 | 0.16 |
| Max | 0.83 | 15.5 | 0.94 | 0.58 | 1050 | 10.9 |
| Mean | 0.27 | 2.9 | 0.83 | 0.31 | 360 | 3.4 |
| 1σ | 0.11 | 2.8 | 0.06 | 0.11 | 178 | 1.83 |
| Median | 0.26 | 1.87 | 0.84 | 0.31 | 350 | 3 |
| DI $n =$ | 85 | 85 | 85 | 85 | 87 | 17 |
| Min | 0.04 | 0.21 | 0.65 | 0.05 | 68 | 0.86 |
| Max | 0.84 | 14.7 | 0.92 | 0.45 | 560 | 2.8 |
| Mean | 0.43 | 1.57 | 0.78 | 0.26 | 320 | 1.47 |
| 1σ | 0.2 | 2.1 | 0.06 | 0.11 | 114 | 0.63 |
| Median | 0.41 | 1.17 | 0.78 | 0.27 | 320 | 1.2 |
| Means different?[a] | Yes | Yes | Yes | No | Yes | Yes |

| | Garnet | | | | | |
| --- | --- | --- | --- | --- | --- | --- |
| | $TiO_2$ | $Al_2O_3/FeO^t$ | $Na_2O$ | Mg# | Ca# | V |
| Unit | wt.% | wt.% | wt.% | | | ppm |
| Xenoliths $n =$ | 258 | 259 | 255 | 259 | 259 | 261 |
| Min | 0.02 | 0.83 | 0.01 | 0.25 | 0.06 | 19 |
| Max | 0.99 | 3.4 | 0.24 | 0.81 | 0.51 | 920 |
| Mean | 0.22 | 1.6 | 0.08 | 0.61 | 0.2 | 141 |
| 1σ | 0.14 | 0.45 | 0.04 | 0.1 | 0.09 | 123 |
| Median | 0.18 | 1.49 | 0.07 | 0.63 | 0.19 | 110 |
| DI $n =$ | 135 | 135 | 135 | 135 | 135 | 134 |
| Min | 0.12 | 0.91 | 0.03 | 0.39 | 0.05 | 95 |
| Max | 1.7 | 2.7 | 0.54 | 0.81 | 0.58 | 560 |
| Mean | 0.63 | 1.45 | 0.2 | 0.57 | 0.24 | 230 |
| 1σ | 0.30 | 0.35 | 0.12 | 0.09 | 0.11 | 99 |
| Median | 0.56 | 1.38 | 0.17 | 0.56 | 0.23 | 220 |
| Means different?[a] | Yes | Yes | Yes | Yes | Yes | Yes |

| | Reconstructed whole rocks[b] | | | | Oxythermobarometry[c] | |
| --- | --- | --- | --- | --- | --- | --- |
| | $Al_2O_3/FeO^t$ | MgO | V | Eu/Eu* | Log $fO_2$ | T |
| Unit | wt.% basis | wt.% | ppm | | ΔFMQ | °C |
| Xenoliths $n =$ | 258 | 258 | 260 | 258 | 59 | 248 |
| Min | 0.84 | 5.5 | 35 | 0.69 | −4.99 | 715 |
| Max | 4.3 | 18.6 | 670 | 2.70 | −1.67 | 1409 |
| Mean | 1.74 | 12.4 | 250 | 1.14 | −3.48 | 1079 |
| 1σ | 0.6 | 2.6 | 115 | 0.29 | 0.81 | 153 |
| Median | 1.57 | 12.2 | 230 | 1.07 | −3.50 | 1096 |
| DI $n =$ | 18 | 18 | 17 | 5 | | 16 |
| Min | 0.87 | 8.1 | 95 | 0.94 | | 969 |
| Max | 2.5 | 16 | 480 | 1.15 | | 1473 |
| Mean | 1.32 | 12 | 270 | 1.03 | | 1188 |
| 1σ | 0.38 | 2.2 | 105 | 0.09 | | 138 |
| Median | 1.32 | 11.9 | 250 | 0.99 | | 1154 |
| Means different?[a] | Yes | No | No | Yes | Na | Yes |

*Mg#* Mg/(Mg + Fe$^{total}$), *Jd* jadeite component in clinopyroxene (cpx), *Ca#* Ca/(Ca + Mg + Fe$^{total}$ + Mn) in garnet (gt), *D* distribution coefficient, *Eu/Eu** chondrite-normalised Eu/(Sm*Gd)$^{0.5}$ (chondrite of ref. [69]).
[a]Based on two-tailed *t*-test for null hypothesis that the means of two populations are equal, and imposing alpha = 0.05 (below which the null hypothesis is rejected), using either equal or unequal variances depending on F-test outcomes
[b]Bulk rocks are reconstructed with 0.55 garnet, 0.45 clinopyroxene minus half the weight of rutile each (e.g., for 1 wt.% rutile 0.545 garnet, 0.445 clinopyroxene); for V reconstruction, estimated rutile modes and concentrations in rutile were considered.
[c]Oxygen fugacities as reported or recalculated relative to the Fayalite-Magnetite-Quartz (FMQ) buffer using the eclogite oxybarometer of ref. [70]; temperatures are derived using the thermometer of ref. [51] in iterative solution with regional conductive model geotherms (see ref. [21]).

**Table 2 Pearson correlation coefficents and statistical significance test for various variables discussed in the text (references in Supplementary Data 1).**

| Suite | $D(V)_{cpx\text{-}gt}$-$T$ (°C) | | | Gt $TiO_2$ (wt.%)–$T$ (°C) | | | Gt V (ppm)–Ca#[a] | | | Gt $Na_2O$ (wt.%)–$T$ (°C) | | |
|---|---|---|---|---|---|---|---|---|---|---|---|---|
| | r | n | p | r | n | p | r | n | p | r | n | p |
| 1 | 0.61 | 28 | 0.001 | 0.91 | 28 | <0.001 | 0.11 | 34 | 0.543 | 0.90 | 24 | <0.001 |
| 2 | 0.09 | 149 | 0.260 | 0.40 | 149 | <0.001 | 0.15 | 151 | 0.067 | 0.26 | 149 | <0.001 |
| 3 | 0.74 | 16 | 0.001 | 0.95 | 16 | <0.001 | 0.92 | 21 | <0.001 | 0.94 | 16 | <0.001 |
| 4 | 0.16 | 17 | 0.541 | 0.51 | 17 | 0.034 | 0.46 | 20 | 0.041 | 0.51 | 17 | 0.036 |
| 5 | 0.78 | 23 | <0.001 | 0.82 | 23 | <0.001 | 0.38 | 24 | 0.067 | 0.69 | 23 | <0.001 |
| 6 | 0.28 | 15 | 0.318 | 0.81 | 15 | <0.001 | 0.19 | 15 | 0.488 | 0.26 | 15 | 0.346 |

| Suite | Gt V (ppm)–$TiO_2$ (wt.%) | | | Gt V (ppm)–$Na_2O$ (wt.%) | | | Gt $Na_2O$–$TiO_2$ (wt.%) | | |
|---|---|---|---|---|---|---|---|---|---|
| | r | n | p | r | n | p | r | n | p |
| 1 | 0.88 | 61 | <0.001 | 0.42 | 57 | 0.001 | 0.66 | 61 | <0.001 |
| 2 | 0.53 | 181 | <0.001 | 0.45 | 182 | <0.001 | 0.93 | 181 | <0.001 |
| 3 | 0.67 | 61 | <0.001 | 0.57 | 61 | <0.001 | 0.86 | 61 | <0.001 |
| 4 | 0.18 | 26 | 0.385 | 0.32 | 26 | 0.106 | 0.76 | 27 | <0.001 |
| 5 | 0.48 | 41 | 0.001 | 0.27 | 41 | 0.083 | 0.74 | 41 | <0.001 |
| 6 | 0.41 | 22 | 0.059 | 0.47 | 22 | 0.027 | 0.8 | 22 | <0.001 |

| | $D(V)_{cpx\text{-}gt}$-$fO_2$ | | | Cpx V (ppm)–$fO_2$ | | | Gt V (ppm)–$fO_2$ | | | WR V (ppm)–$fO_2$ | | |
|---|---|---|---|---|---|---|---|---|---|---|---|---|
| | r | n | p | r | n | p | r | n | p | r | n | p |
| All | 0.34 | 59 | 0.008 | 0.24 | 59 | 0.070 | 0.48 | 59 | <0.001 | 0.33 | 59 | 0.011 |

Suites: 1—Kaapvaal craton kimberlite-hosted, 2—Kaapvaal craton orangeite-hosted, 3—Zimbabwe craton, 4—Northern Slave craton, 5—Central Slave craton, 6—Superior craton.
r—Pearson correlation coefficient, n—number of observations, p—p-value for statistical signficance testing, whereby a value of ≤0.05 is taken to indicate statistical significance (i.e., the null hypothesis of no significant correlation is rejected).
Cpx clinopyroxene, Gt garnet, Ca# molar Ca/(Ca + Mg + Fe^{total} + Mn), D distribution coefficient, oxygen fugacity values $fO_2$ and temperatures as in Table 1 .
[a]Xenoliths only to minimise superposition of temperature effects.

(Supplementary Fig. 1c–d). In contrast, rutile solubility into clinopyroxene with increasing temperature is weaker[25], and the difference between median $TiO_2$ contents in DI clinopyroxene (0.41 wt.%) and xenolithic clinopyroxene (0.26 wt.%) is correspondingly smaller (Fig. 1f; Table 2). With a median estimated rutile V abundance of 1240 ppm in xenoliths (Supplementary Data 1), the amount of V added to garnet if all rutile (median 0.25 wt.%) were completely dissolved in 55 wt.% garnet (the mode used in bulk rock reconstruction; see Methods) would be ~6 ppm. This alone cannot account for the difference between DI and xenolithic garnet, and therefore crystal-chemical effects in addition to rutile-solubility effects are required.

Higher CaO contents in garnet, here gauged by its molar Ca# (Ca/(Mg + Fe^{total} + Ca + Mn)), facilitate incorporation of incompatible trace elements[21,25,27]. Consideration of xenoliths only, to minimise superposition of effects from higher average temperatures recorded for DI, reveals a significant negative correlation of garnet V abundance with Ca# for two of six sample suites (Table 2), which translates into a negative correlation of $D(V)_{cpx\text{-}gt}$ with garnet Ca# (Supplementary Fig. 1e–f). Finally, $Na_2O$ uptake in garnet is temperature-dependent[25], as borne out by significant differences in $Na_2O$ content for garnet in xenoliths and DI (Fig. 1g), and by positive correlations between $Na_2O$ content and temperature for five of six sample suites (Supplementary Fig. 1g–h; Table 2). High V abundances in garnet are correlated to high $TiO_2$ and $Na_2O$ contents (significant for four of six suites each), whereby groups of samples deviate from this pattern (Supplementary Fig. 1i–l; Table 2). The temperature effect may be indirect, via increased uptake of $TiO_2$ and $Na_2O$ contents in garnet, which might enhance V abundances as part of various coupled substitutions.

**Oxygen fugacity effects on V partitioning in eclogitic minerals.** In contrast to homovalent trace elements, $D(V)_{cpx\text{-}gt}$ may also depend on $fO_2$, as it does for mineral-melt pairs because $fO_2$ determines the average valence state and cation size, which in turn determines the fit into various lattice sites available in minerals[28–31]. For further insights into $fO_2$ effects on $D(V)$, we consider both experimental and natural mineral pairs. The combination of parameterisations of experimental clinopyroxene-melt and rutile-melt $D(V)$[12,26] as a function of $fO_2$ (both obtained at 1300 °C) reveals a marked increase in clinopyroxene/rutile $D(V)$ with decreasing $fO_2$, relative to the Fayalite-Magnetite-Quartz buffer expressed as $\Delta\log fO_2$(FMQ), from 0.20 at FMQ-1 to 1.9 at FMQ-5. This roughly corresponds to the $fO_2$ range in eclogite xenoliths in this study (Fig. 2a; Supplementary Data 1). In contrast, $D(V)_{cpx\text{-}gt}$ in eclogite xenoliths shows a significant correlation with $fO_2$ when all data are considered (Table 2). While V abundances in clinopyroxene show no dependence on $fO_2$ ($r^2 = 0.06$; $n = 59$; Fig. 2b; Table 2), those in garnet show a stronger positive correlation ($r^2 = 0.23$; $n = 59$; Fig. 2c; Table 2). The scatter likely reflects superposed crystal-chemical controls unrelated to $fO_2$ effects, as discussed above, and various petrologic processes affecting the cratonic eclogite reservoir from its beginnings at mid-ocean ridges to exhumation with the host magma (next sections).

Despite the scatter, the $fO_2$–V relationships in Fig. 2 suggest that an increase in $fO_2$ facilitates V uptake in garnet and rutile, more so than in clinopyroxene. The observed decrease in experimental $D(V)_{cpx\text{-}rutile}$ may indicate a decreasing misfit with respect to the octahedral site occupied by Ti in rutile[32]. It is uncertain whether the $fO_2$ increase is accompanied by an increase in the average V valence state in garnet. Exclusively trivalent V in octahedral coordination is found in $Fe^{3+}$-rich

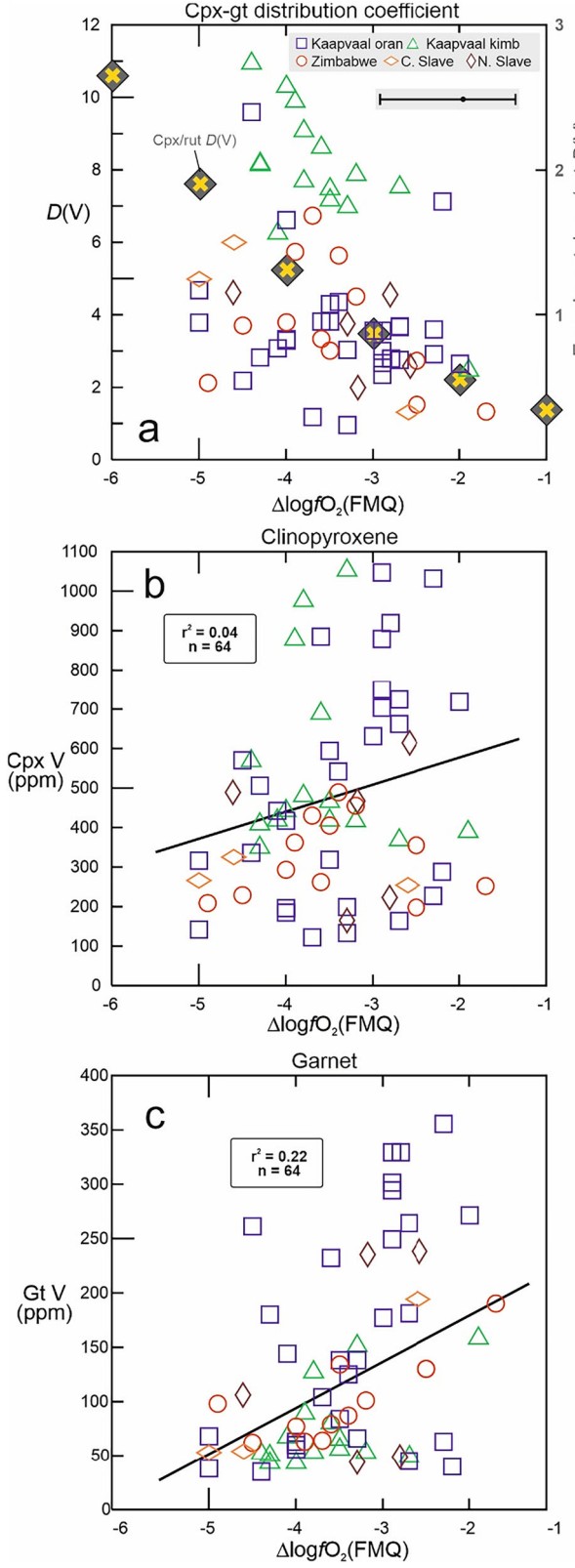

**Fig. 2 Effects of oxygen fugacity on the distribution and abundances of V in eclogite minerals. a** Distribution $D$ of V between clinopyroxene (cpx) and garnet (gt) in eclogite xenoliths as a function of oxygen fugacity relative to the Fayalite-Magnetite-Quartz buffer ($\Delta\log fO_2$(FMQ); available for xenoliths only); oran-orangeite-hosted, kimb-kimberlite-hosted. Superposed is the distribution of V between clinopyroxene and rutile (rut) as a function of $fO_2$ (yellow crosses), as calculated from the parameterisation of experimentally determined mineral-melt partition coefficients[12,26]. Vanadium concentration (ppm) in **b** clinopyroxene and **c** garnet as a function of $fO_2$, with regressions and $r^2$ (significant for the number n of observations; Table 2). Error bar in **a** shows representative average 1σ uncertainty of $fO_2$ estimates of −1.0, +0.6[14]; 1σ uncertainties on V abundances of 14.5 ppm for clinopyroxene and 5.6 ppm for garnet (corresponding to typical uncertainties for multiple analyses per sample reported in the literature, see Methods) are small relative to the depicted scale and not shown. Data sources in Supplementary Data 1.

state of ~+2.5 (i.e., ~50% $V^{2+}$ and 50% $V^{3+}$) over an extended $fO_2$ range between FMQ-4.5 to FMQ-1.8, suggesting that crystal-chemistry exerts a stronger control on V valence in garnet than $fO_2$[29].

**Identifying cumulate vs. melt protoliths.** As mentioned above, eclogite xenoliths are interpreted as metamorphic products of subducted oceanic crust composed of extrusive and intrusive sections[6], and the same applies to Archaean eclogitic diamond substrates[10]. Prior eclogite xenolith work gauged the importance of plagioclase accumulation based on Eu/Eu* (chondrite-normalised Eu/(Sm*Gd)^0.5), which exploits the greater compatibility of Eu than neighbouring REE in plagioclase, and on low abundances of HREE, which are incompatible during accumulation. However, only a few studies report trace-element data for DI. As plagioclase-rich rocks with low HREE abundances, such as MOR gabbros, also have high $Al_2O_3$/FeO (Supplementary Fig. 2), this ratio can serve as an alternative proxy to detect cumulate processes during protolith formation. A plot of V vs. $Al_2O_3$/FeO in reconstructed whole rocks illustrates low V abundances in samples with gabbroic protoliths, and also highlights a higher proportion of such protoliths for xenoliths than for DI (Fig. 3a–b; Table 1). The trend of low V-high $Al_2O_3$/FeO in eclogites with cumulate protoliths to high V-low $Al_2O_3$/FeO in eclogites with melt protoliths mimics that of natural MOR gabbros and MORB, save some reconstructed bulk eclogites from the Kaapvaal and northern Slave cratons showing anomalous V enrichment (Fig. 3a–b). These systematics are also recognisable in clinopyroxene and garnet alone (Supplementary Fig. 2). Thus, lower V in xenolithic garnet than in DI garnet is the result not only of temperature and crystal-chemical effects, as discussed above, but also of greater importance of cumulate processes during protolith formation.

The higher proportion of gabbroic protoliths in eclogite xenoliths than in eclogitic DI is statistically significant (Table 2) and therefore unrelated to sampling issues. Furthermore, a gabbroic signature is also evident for a higher proportion of diamondiferous eclogites than DI[34]. This suggests that it is not the diamond-forming process itself that modifies the eclogite substrate. Instead, in a model of eclogitic diamond formation via flushing of oceanic crust with dehydration fluids derived from the underlying serpentinised oceanic mantle[21], it might reflect that fluids are initially solute-poor as they enter the gabbroic lower oceanic crust, leading to the formation of inclusion-free diamond. One may speculate that such fluids reach carbon saturation in all parts of the oceanic crust but become more solute-rich by the time they infiltrate the upper

garnet of the schorlomite species representing garnet at fairly oxidised conditions[33]. Righter et al.[29] investigated the partitioning of V between garnet and silicate melt at high pressure and found a marked decrease in V compatibility in garnet with increasing $fO_2$. Nevertheless, garnet in the experiments and in natural peridotite was found to have a rather constant average V valence

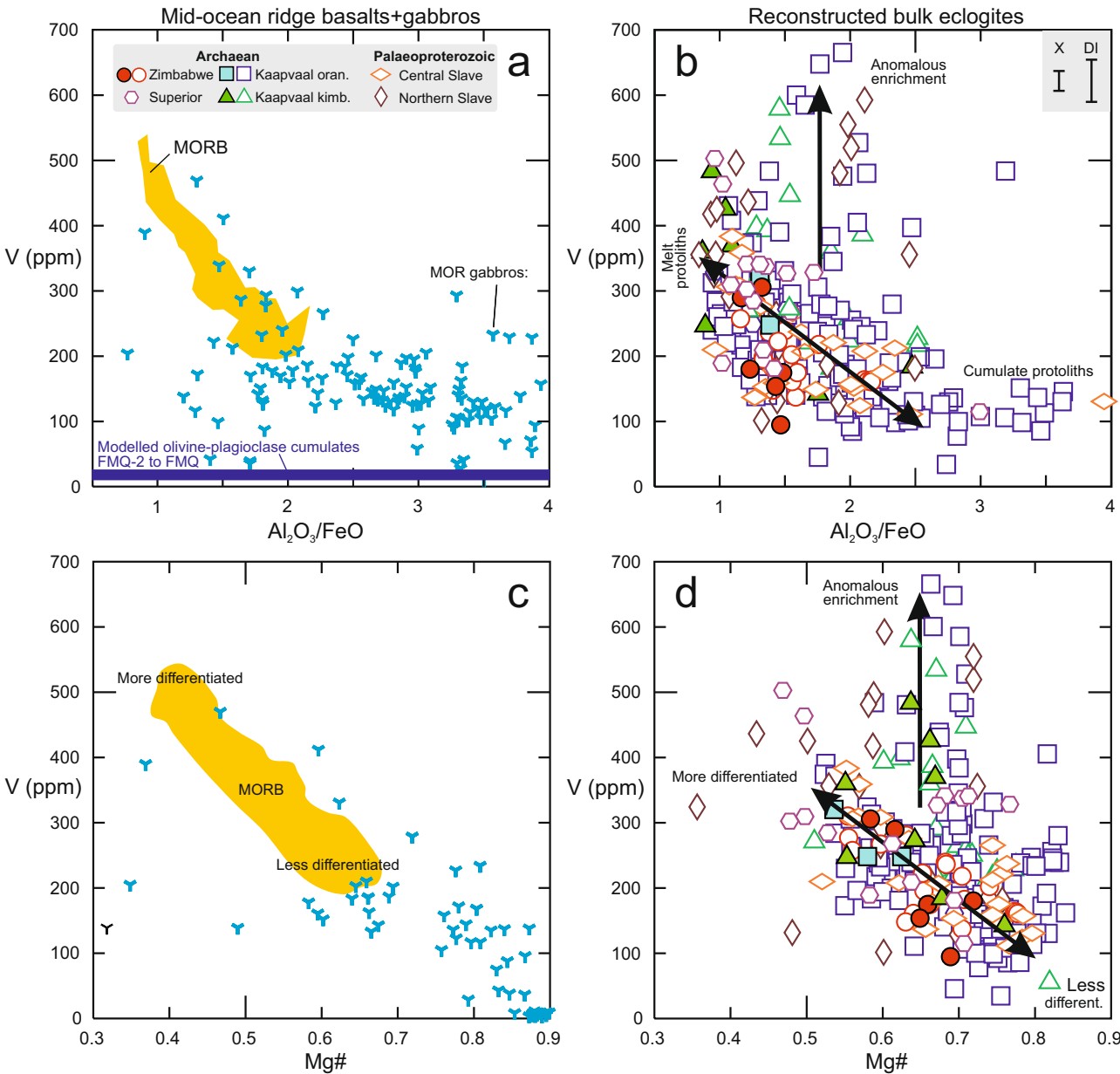

**Fig. 3 Effects of low-pressure accumulation and differentiation on V abundances in eclogite protoliths.** Vanadium abundances (ppm) in **a** mid-ocean ridge basalts (MORB; yellow field[71]) and MOR gabbros (blue tristars, Eu/Eu* > 1.05; from PetDB: www.earthchem.org/petdb), and **b** reconstructed bulk compositions of eclogite xenoliths (X) and inclusions in diamond (DI) as a function of $Al_2O_3$/FeO. Eclogites with suggested gabbroic cumulate vs. melt protoliths, and those showing inferred anomalous enrichment, are indicated in **b**. Shown in **a** is the range of compositions for olivine+plagioclase accumulating from a parental melt formed at $\Delta\log fO_2$(FMQ) −2 to 0 (dark blue bar at bottom; modelling parameters and rationale in Methods; Supplementary Table 2). Vanadium abundances (ppm) in **c** MORB (yellow field) and MOR gabbros (blue tristars), and **d** reconstructed bulk eclogite xenoliths and DI as a function of molar Mg# (Mg/(Mg + Fe$^{total}$)), a proxy for the degree of differentiation during protolith formation. Eclogites with suggested less and more differentiated protoliths, and those showing inferred anomalous enrichment, are indicated in **d**. Error bars for reconstructed bulk rocks in **b** show average propagated 1σ uncertainties on V abundances of 29 ppm for xenoliths and 69 ppm for DI (Methods). Data sources in Supplementary Data 1.

oceanic crust, enabling the precipitation of inclusion-bearing diamonds.

**Effects of petrologic processes on V abundances in eclogite.** In order to assess petrologic effects on V abundances, we model the behaviour of V during the formation and differentiation of oceanic crustal protoliths, as well as during subsequent melting in subduction zones, and metasomatism during their residence in the lithospheric mantle. Mineral-melt $D$(V) during partial

melting of the mantle depends not only on $fO_2$ but also on temperature[35]. Both parameters have changed through geologic time[14,15,36,37], with a suggested lower end of shallow upper mantle $fO_2$ of ~FMQ-2 and mantle potential temperature of $T_P$ ~1450 °C at the time of dominantly Mesoarchaean emplacement of oceanic crust now sampled as eclogite xenoliths. Entering these parameters into the spreadsheet of ref. [35], which uses a Depleted Mantle starting composition with 73 ppm V and calculates mineral-melt $D$(V) as a function of temperature and $fO_2$, the parental melt is modelled to have ~170 ppm V for a melt fraction

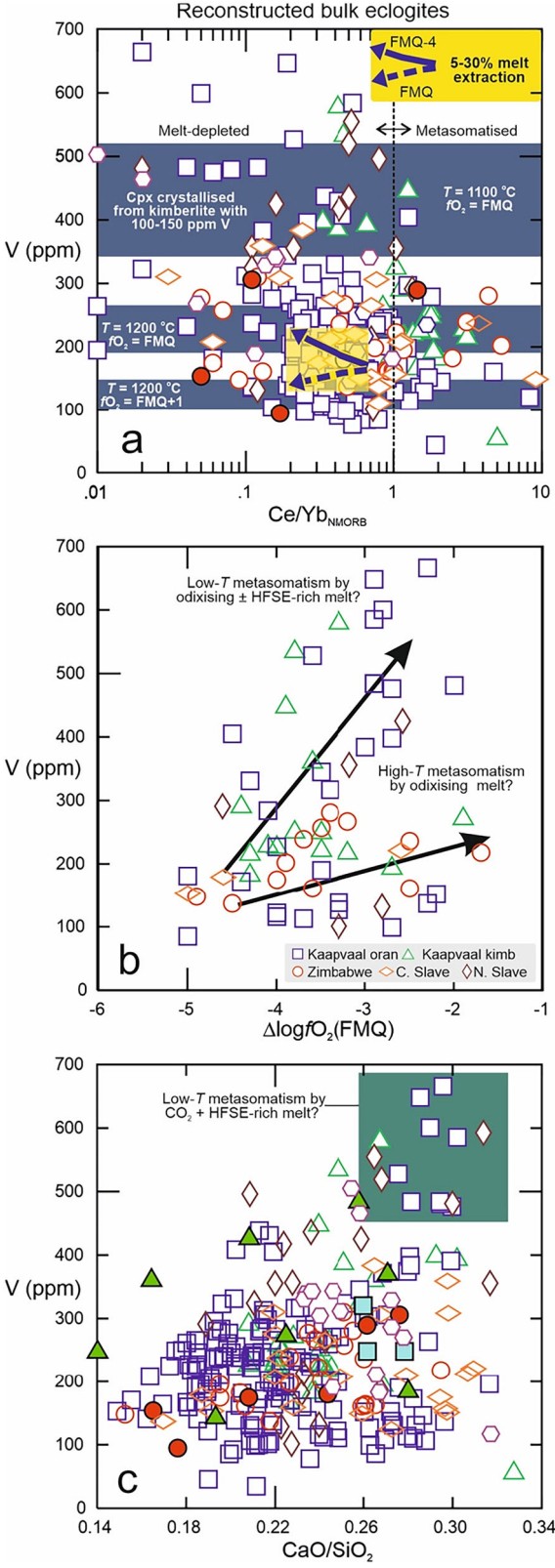

**Fig. 4 Effects of partial melt extraction and metasomatism on V abundances in eclogites. a** Vanadium abundances (ppm) in reconstructed bulk eclogite xenoliths and DI as a function of normal mid-ocean ridge basalt (NMORB)-normalised Ce/Yb, a proxy for melt depletion and enrichment. Shown for comparison is the modelled trend (yellow box) for partial melt extraction from rutile-eclogite, with a moderate decrease of V abundances at relatively oxidising conditions of $\Delta\log fO_2(FMQ) = 0$, similar to the modern ambient mantle[72]. For $\Delta\log fO_2(FMQ) = -4$, at the low end of $fO_2$ estimates for eclogite xenoliths, partial melt extraction from eclogite causes an increase in V abundances (modelling parameters and rationale in Methods; Supplementary Table 4). Also shown is the effect of metasomatism, as mediated by crystallisation of a high-temperature pyroxene from a kimberlite melt[21] with 100–150 ppm V[66]; V abundances in the metasomatic clinopyroxene are calculated for various temperature and $fO_2$ conditions, shown with grey-blue fields (modelling parameters and rationale in Methods; Supplementary Table 5); reconstructed pre-metasomatic eclogites ($Cr_2O_3 < 0.1$ and $Ce/Yb_{NMORB} < 1$) would have $280 \pm 140$ ppm V. NMORB for normalisation from ref. [60]. **b** V abundances (ppm) in reconstructed bulk eclogite xenoliths as a function of oxygen fugacity relative to the Fayalite-Magnetite-Quartz buffer ($\Delta\log fO_2(FMQ)$; no combined $fO_2$-V information available for DI), showing an overall positive correlation (significant, see Table 2). Arrows qualitatively indicate effects of melt metasomatism via clinopyroxene addition under various conditions as shown in **a** (HFSE-high field-strength elements). **c** V abundances (ppm) in reconstructed bulk eclogite xenoliths and DI as a function of $CaO/SiO_2$, a proxy for metasomatism by $CO_2$-rich melt[49]; note samples with anomalous V enrichment are restricted to high ratios (teal-coloured field). Average propagated $1\sigma$ uncertainties on V abundances are 29 ppm for xenoliths and 69 ppm for DI (Methods). Data sources in Supplementary Data 1.

melting for modern compared to Archaean oceanic crust formation.

The crystallisation sequence at pressures <0.5 GPa in dry peridotite-derived melts, typical for spreading ridges, is spinel, olivine, plagioclase and clinopyroxene. So long as only spinel, olivine and plagioclase are on the liquidus (over some 50% of crystallisation), the solid-melt bulk partition coefficient for V is very low (<0.15) for reasonable Archaean conditions ($fO_2 =$ FMQ-2 to FMQ; $T_P = 1450$–$1400 °C$), resulting in a modelled cumulate rock with <30 ppm V (Methods; Supplementary Table 2; Fig. 3a). This is lower than the ~100 ppm for reconstructed eclogite xenoliths with high $Al_2O_3/FeO$ and therefore plagioclase-rich cumulate protoliths (Fig. 3b). Although higher $T_P$ results in higher V in the melt, this is to some extent offset by lower mineral-melt $D(V)$ during crystallisation at correspondingly higher temperatures. The gabbroic protoliths may therefore have experienced the addition of trapped melt and/or reactive crystallisation of clinopyroxene (~110–580 ppm V at the modelled conditions), as postulated for modern MOR gabbros[38]. Still, MOR cumulates have low V, which is then inherited by eclogite upon metamorphism.

Due to the low olivine- and plagioclase-$D(V)$, the fractionation of these minerals during oceanic crust formation leads to an increase in melt V abundances (Methods; Supplementary Table 3). Differentiation along with olivine ± plagioclase control lines during protolith formation may explain why Mg# in garnet, clinopyroxene and whole rocks is anti-correlated with V abundances (Fig. 3c–d, Supplementary Fig. 3). For reconstructed whole rocks, this trend again mimics that observed in modern MORB and MOR gabbros, again with deviations for some samples from the Kaapvaal and northern Slave cratons. The considerable scatter in the natural samples probably reflects the superposition of additional processes, such as partial melt

of 0.2 (see Supplementary Table 1 and Methods for modelling parameters and rationale). For comparison, the parental melt of modern MORB at $F = 0.08$, $T_P = 1300 °C$ and $fO_2 = FMQ$ would have ~200 ppm V. This relatively small difference reflects competing effects of stronger V compatibility in the solid at lower $T_P$ but stronger V incompatibility at higher $fO_2$ during mantle

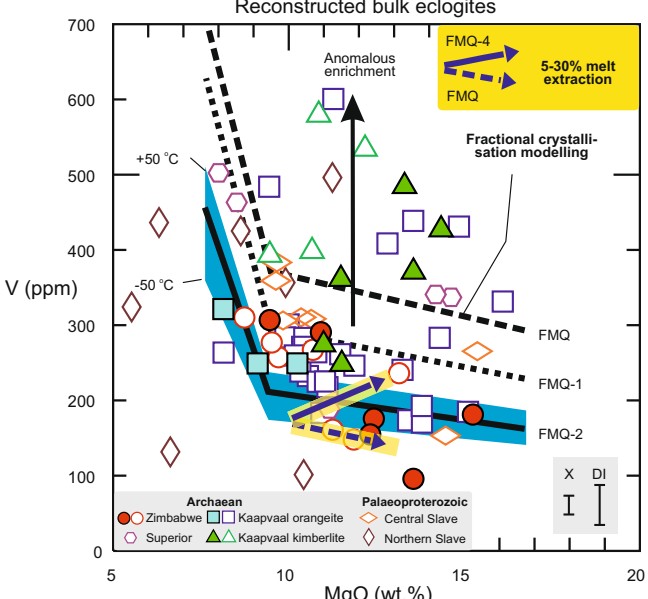

**Fig. 5 Estimates of _f_O₂ of convecting mantle sources from which the oceanic crustal protoliths to eclogites formed.** Shown are V abundances (ppm) in reconstructed bulk eclogite xenoliths and DI as a function of MgO content (wt.%), filtered to exclude samples with gabbroic cumulate protoliths (conservative cut-off Al₂O₃/FeO = 1.7) and, for xenoliths, metasomatism (NMORB-normalised Ce/Yb > 1 and Cr₂O₃ > 0.1 wt.%). Shown for comparison is the modelled evolution of melts for a mantle potential temperature ($T_P$) of 1450 °C (olivine crystallisation temperature of 1340 °C) after crystallisation of spinel, olivine and plagioclase from a parental melt formed at a melt fraction of 0.2, appropriate for warmer Archaean spreading ridges, and various _f_O₂. A difference of ±50 °C in $T_P$ and corresponding crystallisation temperature is shown as blue envelope for Δlog_f_O₂(FMQ) = −2, whereby V becomes more incompatible with increasing temperature[35], causing the isopleths to shift upward relative to samples. Modelled effect of 30% melt extraction at FMQ-4 and FMQ (as in Fig. 4a) is shown for comparison (purple arrows), illustrating that melt extraction at FMQ-4 biases V abundances toward higher apparent _f_O₂ whereas at FMQ melt extraction shifts samples with MgO > ~9 wt.% along olivine control lines. Modelling parameters and rationale in Methods; Supplementary Table 3. Error bars for reconstructed bulk rocks show average propagated 1σ uncertainties on V abundances of 29 ppm for xenoliths and 69 ppm for DI (Methods). Data sources in Supplementary Data 1.

extraction during subduction of oceanic crust, and metasomatism after its capture in the lithospheric mantle.

Melt extraction during subduction decreases the LREE/HREE ratio because of the incompatibility of LREE and the compatibility of HREE in a rock consisting of subequal volumes of clinopyroxene and garnet (Fig. 4a). Modelling shows that this process leads to moderately decreasing V abundances in the residue (e.g., from 170 ppm in the protolith to 140 ppm after 30% of melt extraction), at relatively oxidising conditions of _f_O₂ = FMQ, and to insignificant changes (to 176 ppm) for FMQ-1, but increasing V abundances for lower _f_O₂ (to 230 ppm for FMQ-4) (Fig. 4a; Methods; Supplementary Table 4). The modelled changes due to partial melt extraction are small relative to the range displayed by natural samples and therefore do not appear to be the dominant control on V abundances in eclogite xenoliths or DI.

It has been inferred that the compositional relationships in metasomatised eclogite xenoliths are best satisfied if magmatic

clinopyroxene is added from a kimberlite-like melt[21]. The parameterisation of ref. [35] demonstrates that V concentrations in clinopyroxene in equilibrium with melt would vary strongly not only as a function of _f_O₂ but also of temperature. Modelling shows that the addition of such clinopyroxene would most strongly increase V abundances in eclogite for reducing and cool conditions when V behaves most compatibly (Fig. 4a; Methods; Supplementary Table 5).

## Discussion

Earlier work suggested generally more reducing conditions in the Archaean convecting mantle than today. This is based on low V/Sc and Fe³⁺/ΣFe in unmetasomatised, non-cumulate and little-differentiated (meaning only olivine ± plagioclase fractionation) eclogite specimens[14], and on _f_O₂ estimates from komatiites, picrites and basalts of various ages[15,39]. When the effects of accumulation and fractional crystallisation during protolith formation are accounted for, by considering V abundances as a function of Al₂O₃/FeO and Mg#, respectively, there are no systematic differences between xenoliths and DI (Fig. 3). Instead, eclogite compositions reconstructed from both xenoliths and DI predominantly follow trends recognised also in modern MORB and MOR gabbro, although a subset shows anomalous enrichment that is discussed below. In Fig. 5, all reconstructed bulk eclogites were filtered for accumulation, and xenoliths additionally for metasomatism, for a meaningful comparison to forward models of differentiating mantle-derived melts. Most of the Archaean data fall between modelled _f_O₂ isopleths of ~FMQ-1 to FMQ-2 (Fig. 5). In contrast, Palaeoproterozoic eclogites from the central Slave craton mostly fall between ~FMQ-1 to FMQ, while those from the northern Slave craton show significant scatter (see next section).

Melt loss from eclogite at FMQ would cause a decrease in V abundance that parallels that of olivine control lines, causing little change in apparent source _f_O₂ (Fig. 5). Conversely, melt loss from eclogite at _f_O₂ ≤ FMQ-1, corresponding to those determined for the vast majority of samples, would increase V abundances, biasing the results towards a higher apparent source _f_O₂. This suggests that in carefully filtered databases lacking information on geochemically similar homovalent elements (e.g., Sc), V alone can be used to illustrate redox-related igneous processes and capture the maximum _f_O₂ signature of the convecting mantle source from which the igneous protoliths were derived.

In cratonic eclogite, mantle metasomatism, typically by kimberlite-like melts, often causes Cr₂O₃ and LREE enrichment (e.g., NMORB-normalised Ce/Yb > 1), and is thought to be mediated by the precipitation of a high-temperature pyroxene[21]. Mantle metasomatism is often associated with oxidation[40,41], although both oxidising and reducing metasomatism occurs, based on garnet oxybarometry applied to peridotite xenoliths[42]. For oxidising metasomatism by carbonated melts, a corresponding decrease of V abundances in olivine and two pyroxenes from garnet-free peridotite xenoliths has been documented[43]. This is consistent with experiments, where peridotite minerals show a strong decrease in mineral-melt D(V) with increasing _f_O₂ across the _f_O₂ range relevant to the cratonic mantle[12]. In contrast, this study makes the intriguing observation that with increasing _f_O₂, V abundances in eclogite garnet (Fig. 2b, c) and bulk rocks (Fig. 4b) from some suites show a mild but statistically significant increase (Table 2) or remain constant for others.

The seemingly paradoxical effect of increasing V abundances with _f_O₂ in some xenolithic eclogite suites may result from the stability of low-volume, carbonated oxidising melts in eclogite to temperatures that are lower by several 100 °C than temperatures required for stabilisation of higher-volume, carbonate-poor and

therefore less oxidising melts[44]. Vanadium concentrations in pyroxene precipitated from a kimberlite melt with 100–150 ppm V under cool conditions (1100 °C) would be V-rich even at relatively high $fO_2$ of FMQ (e.g., 350–530 ppm), but much lower for higher melt-rock reaction temperatures of 1200 °C (180–280 ppm; Fig. 4a). For comparison, reconstructed pre-metasomatic eclogites, i.e., those having $Cr_2O_3 < 0.1$ and $Ce/Yb_{NMORB} < 1$, have V concentrations of 280 ± 140 ppm. Metasomatic clinopyroxene can therefore increase or dilute V abundances in eclogite, depending not only on $fO_2$ but also on the temperature conditions under which it was added. This may explain why oxidative metasomatism in mantle eclogite causes no consistent change in the V abundance of the affected eclogite, and can even cause an increase in V abundances in some samples (Fig. 4b), in contrast to the decrease observed in peridotite with generally higher solidus temperatures[45].

It is conspicuous that eclogite xenoliths from the northern Slave craton and some samples from the Kaapvaal craton show anomalous enrichment in V, extending to concentrations beyond those in any DI garnet (Fig. 3). Eclogite xenoliths from the Slave craton have been previously noted for their unusual HFSE enrichment during interaction with $CO_2$-rich melt soon after subduction and metamorphism[46], and it seems that V was enriched along with Nb and Zr. In the Kaapvaal craton, anomalous V enrichment may be linked to the formation of MARID rocks (Mica-Amphibole-Rutile-Ilmenite-Diopside) around the time of orangeite magmatism[47,48]. In both cratons, the enrichment would be related to an external, HFSE-enriched melt, which does not appear to impart elevated Ce/Yb (Fig. 4a). Nevertheless, the involvement of a silica-undersaturated, $CO_2$-rich agent is supported by the observation that eclogites with the highest V abundances are restricted to elevated $CaO/SiO_2$, a hallmark of carbonatite metasomatism[49] (Fig. 4c).

Diamond formation is a metasomatic process involving COH fluids or melts[3,10]. However, once formed, a diamond is a robust container for inclusions. In contrast, the mantle as sampled by xenoliths at the time of kimberlite entrainment has remained an open system. In this light, it is somewhat surprising that most reconstructed bulk xenoliths and DI show similar V abundances at similar $Al_2O_3/FeO$ and Mg#, i.e., after accounting for effects of low-pressure accumulation and differentiation (Fig. 3). This retention of low-pressure signatures in both xenoliths and DI attests to the robustness of V during processes after the ocean floor stage. The overall similarity of V abundances in xenoliths and DI may thus point to an oxygen-conserving mechanism for eclogitic diamond formation, for example via mixing, or cooling of $CO_2$- and $CH_4$-bearing fluids near the water maximum[2,3]. The implication is that diamond formation does not appear to alter the $fO_2$ of the eclogites.

The similarity of V abundances in DI and xenoliths further suggests that V concentrations in the cratonic eclogite reservoir globally—and $fO_2$ conditions—did not change systematically subsequent to the diamond formation, despite the scatter in V abundances and $fO_2$ imposed by metasomatism. The implication is that the mantle eclogite reservoir, and by inference subducted oceanic crust, is a system that is efficiently buffered to $O_2$ introduced over aeons of subduction- and mantle-related metasomatic processes. Instead, these processes lead to the formation of diamonds at the low $fO_2$ recorded by eclogite xenoliths. This is what makes eclogite a supreme diamond source rock, in addition to being a faithful recorder of the origin, evolution and fate of ancient subducted ocean basins.

## Methods

**Database**. The database for V concentrations in clinopyroxene and garnet inclusions in diamond (DI) is comprised of 87 clinopyroxenes (18 with REE data) and 135 garnets (29 with REE data, 2 without reported major element data). These diamonds are mostly sampled as kimberlite-borne xenocrysts in diamond processing facilities, rarely from diamondiferous xenoliths (e.g., refs. [9,50]). Of these inclusions, 18 are paired (i.e., separate inclusions in the same diamond that, occasionally, are touching), thus conducive to bulk rock reconstruction and temperature calculation (Supplementary Data 1). Temperatures for clinopyroxene-garnet pairs were obtained using the thermometer of ref. [51], solved iteratively with regional conductive model geotherms (see ref. [21] for rationale and limitations). Here, reported compositions for multiple non-touching inclusions per mineral and diamond were averaged. All of these represent measurements from distinct samples, i.e., reported compositions for multiple inclusions in the same diamond were averaged.

For eclogite xenoliths, the choice of datasets reflects localities for which DI V data are also available, encompassing the Zimbabwe craton, the Kaapvaal craton at the time of Jurassic orangeite and Cretaceous kimberlite magmatism, the Superior craton, and central and northern Slave craton (Supplementary Data 1). Only xenolith datasets with major and trace-element analyses for both clinopyroxene and garnet are considered. Samples showing evidence for kimberlite contamination were filtered (here: Ba or Nb in garnet >0.5 ppm, or Ba or Nb in cpx > 1.5 and 1 ppm, respectively)[21]. All data points represent measurements from distinct samples.

**Assessment of analytical methods and uncertainty estimates**. Vanadium is a multi-valent element, the partitioning of which into minerals vs. melt depends also on its redox state[11,12]. The following assessment of analytical techniques is centred on V, as abundances derived from mixed techniques are used in this study; these are indicated in Supplementary Data 1.

Published $V_2O_3$ concentrations in DI were predominantly determined by EPMA. Limits of detection (LOD) for V are not always given; ref. [52] quote 100 ppm, and ref. [53] quote ≤136 ppm for all measured oxides. We here apply a blanket LOD of 136 ppm assuming that some lower reported values may actually be <LOD. To avoid biasing the DI database toward higher values, we count values <136 ppm as LOD * sqrt(2)/2 (ref. [54]), corresponding to 96 ppm. ref. [52] reports external average 1σ uncertainties for EPMA-derived V concentrations (multiple spots per mineral and sample) of 21% and 52% for clinopyroxene and garnet, respectively, which we assign as a blanket uncertainty to EPMA-derived mineral data for all DI (67 ppm for clinopyroxene, 117 ppm for garnet, for their respective mean compositions in the database). The overlap of the TiKβ on the VKα peak during wavelength-dispersive spectroscopy implies that V concentrations have to be corrected, for example by analysing a Ti-rich and nominally V-free standard as an unknown to apply a linear correction based on apparent V concentrations (e.g., data reported in ref. [55]). Other work (e.g., in ref. [53]) employed the software of ref. [56] with the JEOL 8900 R, which makes an automated correction. All EPMA-derived V concentrations have been corrected for Ti overlap either offline or using the software, as indicated in Supplementary Data 1. Reference [35] considers the correction procedure for $TiO_2$ contents <1.76 wt.% unproblematic. In this study, reported $TiO_2$ abundances in eclogitic clinopyroxene and garnet DI are relatively low (median 0.43 and 0.64 wt.%, respectively) requiring only moderate correction.

Vanadium is a relatively abundant element in clinopyroxene (average 350 ppm in this study) and garnet (140 ppm) from eclogite xenoliths ($n = 261$), far above typical LOD afforded by laser-ablation microprobe inductively-coupled plasma mass spectrometry (LAM-ICPMS; e.g., average LOD 0.05 ppm for >100 analyses reported in ref. [21]). Reference materials were measured as unknowns to monitor accuracy in all xenolith studies included in Supplementary Data 1, and no issues are reported (two DI studies with LAM-derived V abundances do not comment on accuracy). Where available, measured and accepted values for reference materials are given in Supplementary Data 1. Furthermore, the typically high degree of inter- and intra-grain homogeneity affords analysis at high precision (e.g., 4% instrumental, 4% for analysis of multiple spots per grain and multiple grains per sample[21]). For LAM-ICPMS-derived V concentrations we assume a blanket external uncertainty of 4% 1σ (14.5 ppm for clinopyroxene, 5.6 ppm for garnet, for their respective mean compositions in the database).

In order to test the correspondence of LAM- and, after correction, EPMA-derived V abundances, we carried out $V_2O_3$ analyses at Goethe-University Frankfurt employing the Jeol JXA-8530F Plus Hyperprobe, for which V abundances had been determined by LAM-ICPMS. For this, we chose three mounts from Orapa, which are also used in this study (OE16, OE23, OE34; Supplementary Data 1). The results are reported in Supplementary Table 6. A nominally V-free synthetic rutile standard was used to determine the apparent V abundances resulting from the overlap of TiKβ on VKα, using PETL and LIFL crystals, respectively, with average detection limits of 200 ppm (40 s on peak–20 s on background) and 160 ppm (90–45 s), respectively. Apparent V abundances in the synthetic rutile amounted to 2660 ± 68 ppm ($n = 13$), i.e., 26.7 ± 0.7 ppm per wt.% $TiO_2$, which corresponds to 4% of the total V determined in clinopyroxene, and to 11% in garnet. Compared to LAM-derived V abundances, corrected EPMA-derived abundances determined for clinopyroxene in three samples are higher or lower by ≤5% (i.e., without a consistent sign). Corrected EPMA-derived V abundances for garnet in two samples (garnet was not mounted in the third sample) are near and below the detection limit, respectively, with large resultant

uncertainties (190 ± 50 and 110 ± 90 ppm, respectively). They are higher by 3 and 12%, respectively, than LAM-ICPMS-derived concentrations but agree with the uncertainties.

**Bulk rock reconstruction and effect of rutile**. Conventionally, bulk rocks for xenoliths are reconstructed from mineral modes to avoid the effects of kimberlite contamination[6], whereas for DI this is required by the nature of the sample. For both types of samples, the same approach is used, assuming 55 wt% garnet and 45 wt% clinopyroxene, as rationalised in ref. [57]. In the database, the average increase in V related to an estimated 10% uncertainty in the modal proportions of clinopyroxene at the expense of garnet is 21 ppm, or 9%. Consistent with rutile-melt distribution coefficients (D) > 1 (ref. [26]), bulk rocks reconstructed without rutile have minimum V contents. The modal fraction of accessory rutile in a bulk rock having no primary Ti anomaly, as applies to crystallisation from dry peridotite-derived melt in spreading ridges, can be estimated by assuming that Ti is not depleted relative to REE with similar bulk distribution coefficients during partial melting (Sm and Gd)[57]. As the vast majority of clinopyroxene and garnet in the DI database are unpaired, bulk rocks can be reconstructed only for a small subset of samples, and the theoretical amount of rutile in the source cannot be calculated except for the few samples with reported REE concentrations.

The abundances of V in rutile can be estimated from the abundances measured in clinopyroxene and using a distribution coefficient $D$(V) between clinopyroxene and rutile, assuming these two minerals were in equilibrium. Based on natural clinopyroxene-rutile pairs in 22 eclogite xenoliths, $D$(V) is 0.283 ± 0.091[25], which is biased towards available data from the West African and central Slave craton. For samples with independent $f$O₂ estimates (n = 59 in this study), experimental cpx/rutile distribution coefficients can alternatively be used to estimate V abundances in rutile, taking into account the $f$O₂-dependent partitioning of V in both minerals[12,26] (Supplementary Data 1). Bulk rocks (n = 50) reconstructed with rutile with a fixed clinopyroxene-rutile $D$(V) and, alternatively, with $f$O₂-dependent $D$ have average V concentrations of 309 and 303 ppm, respectively, which can be considered identical within uncertainties. Since only a subset of samples has $f$O₂ estimates, we proceed with rutile V estimates from the average clinopyroxene-rutile $D$(V) determined for eclogite xenoliths.

Uncertainties on V abundances are estimated by propagating the average 1σ uncertainties on V abundances in garnet and clinopyroxene (52% and 21%, respectively, for DI, 4% for both minerals in xenoliths, see above), weighted by their respective contributions (not considering rutile), and considering a total 10% uncertainty in modal abundances. Clinopyroxene contributes 68% to the bulk in xenoliths, but only 54% to the bulk in DI owing to stronger V partitioning into garnet at higher average temperatures recorded by DI, as discussed in Section Temperature and crystal-chemical effects on V distribution. This results in average 1σ uncertainties of 29 ppm for reconstructed bulk xenoliths and 69 ppm for DI.

**Statistics**. Two-tailed $t$-tests were applied to test the null hypothesis that the means of the two populations (xenoliths and DI, reported in Table 1 along with number n of observations) are equal, using either equal or unequal variances depending on F-test outcomes, where an alpha value of 0.05 was imposed. The significance of correlations between various variables discussed in the text was assessed using the $p$-value, whereby a value of ≤0.05 is taken to indicate statistical significance (i.e., the null hypothesis of no significant correlation is rejected); the corresponding results along with Pearson correlation coefficients r and number of observations are reported in Table 2. All statistical assessment was carried out using Microsoft Excel software. No Bayesian analysis or hierarchical and complex designs were performed.

**Rationale and parameters for trace-element modelling**. Model inputs and outputs described below are presented in Supplementary Tables 1–5.

Parental melt composition ($C_0$). To model V concentrations in crystals accumulated from a melt, an estimate for the parental melt composition is needed. The partitioning of V between olivine, the pyroxenes, spinel and melt as a function of temperature and oxygen fugacity has been parameterised by ref. [35] to predict concentrations in peridotite-derived partial melts at different melt fractions. We use their spreadsheet to find V abundances in the melt at conditions specified in Supplementary Table 1. Wang et al.[35] employ a Depleted Mantle starting concentration of 73 ppm, which is adopted, consistent with findings that the mantle source of the protoliths to eclogite xenoliths was depleted (if heterogeneous) by ca. 3.0 Ga ago[36], the age of the oldest suite. The Mesoarchaean ambient convecting mantle, from which the oceanic crustal protoliths of the majority of eclogite xenoliths were formed[7,17], was more reducing than today by 1–2 log units[14,15,39] and hotter by some controversial amount[36,37,58]. Here, we assume an Archaean mantle potential temperature $T_P$ of ~1450 °C, some 150 °C hotter than today[36]. These conditions imply higher melt fractions than today[59] and also have an effect on bulk distribution coefficients during peridotite melting[35]. For an assumed melt fraction of 0.2 (ref. [36]) and Δlog$f$O₂(FMQ) of −2, the peridotite-derived aggregated partial melt would contain ~170 ppm V; the same melt would contain ~300 ppm V for $f$O₂ = FMQ (Supplementary Table 1). For comparison, a parental melt formed at a fraction of 0.08, $f$O₂ = FMQ and $T_P$ of 1300 °C, taken to be representative of conditions of modern MORB formation, would contain ~210 ppm V. The parental MORB V estimate is lower than that of modern average NMORB (280 ppm) because the latter represents a differentiated magma with 7.8 wt.% MgO[60].

Low-pressure accumulation. We estimate the expected concentrations of V in a spinel-olivine-plagioclase cumulate for batch crystallisation using the $C_0$ estimate of 170 ppm for Δlog$f$O₂(FMQ) of −2 and 300 ppm for $f$O₂ = FMQ described in the previous section. The corresponding $f$O₂ and crystallisation temperature (1340 °C; converted from $T_P$ of 1450 °C following ref. [59]) were entered into the spreadsheet of ref. [35] (all other parameters were adopted from the spreadsheet; note that the parameterisation of $D$(V) does not contain a pressure term), wherefrom the temperature- and $f$O₂-dependent olivine- and spinel-melt $D$ were retrieved. As there are no published $f$O₂-dependent $D$(V) values for plagioclase, the average olivine/plagioclase-$D$(V) = 2.5[61–64] is used to estimate plagioclase-$D$(V) from that of olivine. For simplicity, the cooling of the melt during progressive crystallisation is ignored. Accumulation of 1% spinel, 20% olivine and 28% plagioclase prior to the onset of clinopyroxene crystallisation is modelled, based on the results of thermodynamic modelling for fractional crystallisation of a picritic melt at FMQ and a pressure of 0.05 GPa (from Appendix 5 in ref. [57], neglecting small differences in phase relations arising from differences in $f$O₂ (see example modelled in Appendix 5 of ref. [57]). The modelled cumulate has <30 ppm V at all investigated conditions (Supplementary Table 2).

Low-pressure melt evolution. The effect of progressive fractionation of 1% spinel, 20% olivine and 28% plagioclase (assumed to be sequential for simplicity) on V abundances in the melt is modelled using the $C_0$ estimate of 170 ppm at FMQ-2, and alternatively 300 ppm V at $f$O₂ = FMQ, $D$ values derived as described above, and the fractional crystallisation equation. At the chosen crystallisation temperature of 1340 °C, V switches from behaving compatibly in clinopyroxene at FMQ-2 to mildly incompatibly at FMQ, whereas it is always compatible in spinel, and incompatible in olivine and plagioclase (Supplementary Table 3). A difference of ±50 °C in $T_P$ and corresponding crystallisation temperature is additionally explored for $f$O₂ = FMQ-2, illustrating that V becomes more incompatible with increasing temperature[35]. Thus, if higher $T_P$ is assumed for the Archaean, the $f$O₂ isopleths shown in Fig. 5 are shifted upward relative to the samples, implying even more reducing conditions for Archaean oceanic crust.

High-pressure partial melt extraction (metamorphism). Oceanic crust subducting in the Archaean plausibly lost a partial melt[9,65,66]. This is modelled as melt extraction from a rutile-eclogite[67], using garnet-melt and clinopyroxene-melt $D$ from ref. [68] for REE. For V, rutile-melt $D$ as a function of $f$O₂ was parameterised from ref. [26], clinopyroxene-melt $D$ as a function of $f$O₂ was parameterised from ref. [12] and garnet-melt $D$ values were assumed to be identical to clinopyroxene-melt $D$ based on results reported in ref. [12]. These coefficients were chosen because the distribution coefficients suggested by ref. [35] for peridotite melting do not return realistic clinopyroxene-melt $D$(V) for the clinopyroxene compositions and relatively low temperatures near the eclogite solidus. While ref. [68] does report clinopyroxene- and garnet-melt $D$(V), their experiments were carried out at estimated $f$O₂ near IW, which is lower than $f$O₂ estimates for most eclogite xenoliths. Following ref. [57], the residue composition was calculated for weight fractions of 0.445 clinopyroxene, 0.55 garnet and 0.005 rutile at various $f$O₂, using a V concentration of 170 ppm as an eclogite starting composition. Whereas V remains compatible in rutile at all conditions, it behaves compatibly in clinopyroxene and garnet up to FMQ-1, but mildly incompatibly at FMQ (Supplementary Table 4). These results are semi-quantitative in the absence of experimental constraints on the $f$O₂-dependent partitioning of V between eclogite garnet, clinopyroxene and melt.

Metasomatic clinopyroxene addition. The V concentration in clinopyroxene added by kimberlite metasomatism[21] is modelled for crystallisation temperatures of 1200 and 1100 °C as examples (i.e., >200–300 °C hotter than the average residence temperature of eclogite xenoliths; Table 1), and using distribution coefficients parameterised by ref. [35], as above. This assumes that the $f$O₂-dependent clinopyroxene-melt partitioning also applies to carbonated silicate melts, such as kimberlites, for which redox-dependent V partitioning data are not yet available. Vanadium concentrations in kimberlites vary considerably (e.g., 96 ± 62 ppm for the Renard kimberlite, 149 ± 32 ppm for the Wemindji kimberlite, both in the Superior craton[66]), and this is the range explored in the modelling. For kimberlite with 100–150 ppm V, $f$O₂ = FMQ and crystallisation temperature of 1100 °C, the resulting clinopyroxene would have 350–530 ppm V, but only 180–280 ppm V if the temperature were 1200 °C for $f$O₂ = FMQ and 100–150 ppm V if the temperature were 1200 °C for $f$O₂ = FMQ + 1.

## Data availability

No new data were generated as part of this study. References for the data on which this study is based are provided in the Supplementary Information file Supplementary Data 1.

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

## Acknowledgements
Andrew Locock at University of Alberta is thanked for discussion of sources of error when determining V abundances by electron probe microanalyser (EPMA), and Dominik Hezel at Goethe-University Frankfurt for his help with testing the accuracy of $V_2O_3$ determination by EPMA. The Deutsche Forschungsgemeinschaft is gratefully acknowledged for funding to S.A. (DFG-grant # AU356/11).

## Author contributions
S.A. conceived the study, collated xenolith data and wrote the first draft of the manuscript. T.S. collated the DI data and contributed to discussion and manuscript revision.

## Funding

## Competing interests
The authors declare no competing interests.
