## [Peer Review File · Nature Communications]

REVIEWER COMMENTS

Reviewer #1 (Remarks to the Author):

In short:

I recommend that this manuscript undergo substantial revision, and because the authors' conclusions may change, that any revised versions of this manuscript undergo peer review.

Aulbach and Stachel collate and filter a global dataset to compare concentrations of V in garnet and cpx in eclogitic xenoliths (X) with inclusions of these minerals in diamonds (DI) from the same locality. This comparison is useful for examining the effect of metasomatism and melt extraction on the V composition of eclogitic minerals because diamond inclusions are shielded from these processes post-encapsulation. The authors find the mean concentrations of V in DI garnets are greater than in X garnets from the same locality, with the increase facilitated by coupled substitution of V with Na in DI garnets. The authors reconstruct the bulk rock compositions from cpx and garnet data to assess the impact of low-pressure differentiation (i.e., melting at MOR) and partial melt extraction and metasomatism at high pressure on the V contents of eclogites. A key conclusion of this manuscript is that DI garnets appear to contain oxidized species of V (V^{4+}) despite their inferred low oxygen fugacities.

The general premise of this manuscript is sound and the dataset comparing DI to X eclogitic minerals will be of interest to many scientists. However, the authors' conclusions are overstated given the broad scatter and lack of coherent trends in their dataset. Some conclusions, such as the presence of V^{4+} in DI garnets, are not supported by this data presented in this study or by literature data. **Importantly, the DI dataset has not been uniformly corrected for EPMA Ti-V overlap and as such cannot be used for comparison, modeling and interpretation.** I recommend that this manuscript undergo substantial revision, and because the authors' conclusions may change, that any revised versions of this manuscript undergo peer review. Specific major comments follow.

1. **Ti-V correction of the DI data set.** The diamond inclusion dataset, mostly collected by electron probe microanalysis, has not been uniformly corrected for overlap of the $TiK\beta$ and $VK\alpha$ peaks. The impact of this overlap is that $TiK\beta$ counts will be erroneously attributed to $VK\alpha$ such that the calculated concentrations of V are higher than 'true' values. The effect scales with Ti content—as Ti increases, the overlap worsens and calculated V contents increase. The degree of overlap is also highly dependent on what crystals are used in the EPMA analytical routine because the $TiK\beta$ peak is closer to the $VK\alpha$ peak on LPET crystals than on LLIF crystals. The overlap is about 10x 'worse' with LPET compared to LLIF, with about 0.6% of counts on pure Ti reference materials attributed to V with analysis using LLIF crystals vs about 6% of counts with LPET crystals. Note that it is typically more common to analyze these elements on LPET crystals because they yield higher count rates compared to LLIF. Both elements must be analyzed on the same crystal to apply the overlap correction. Analyzing V on LLIF and Ti on LPET, for example, does not avoid or eliminate the overlap issue. A key conclusion of the authors is that DI garnets contain more V than X garnets, which they attribute to the presence of increased Na in the DI garnets to charge balance V. However, the DI garnets contain on average more Ti than X garnets, raising the possibility that the increase in V is simply due to uncorrected Ti-V overlap in EPMA analyses. Nearly all DI data in this study are from EPMA measurements, but only a small

subset of these data are reported as being corrected for Ti-V overlap (lines 413-417) [note analyses of X garnets using LAM-ICP-MS are not subject to this overlap]. The authors' assumption that TiO₂ contents <1.76 wt% are unproblematic is also not acceptable for the purposes of this study. For example: assuming for simplicity's sake a 1:1 relationship between counts and concentration, analysis of a garnet containing 0.68 wt% TiO₂ (e.g., DI sample 245 from Kaapvaal) would result in 277 ppm of Ti that would be erroneously attributed to V without overlap correction. This is more than enough 'vanadium' to significantly impact the authors results and interpretations.

The potential impact of Ti-V overlap can be seen in the DI data collated in the Supplement. For instance, the dataset of Phillips et al 2004 clearly shows a linear increase in both V in garnet and cpx as TiO₂ content of both minerals increases, as would be expected if a Ti-V correction is not deployed.

In sum: vanadium concentrations from EPMA are meaningless without application of a Ti-V overlap. This manuscript cannot be published containing V data from the EPMA that have not be subject to a Ti-V overlap.

2. **Overstating of conclusions.** Many of the conclusions in this manuscript are overstated and do not adequately reflect the breadth and scatter apparent in the authors dataset. Other conclusions are simply not supported by the literature data. Specific examples:
 - a. **V in garnet comparison.** Setting aside the possible impact of Ti-V overlap on the DI garnet data, an important conclusion of the authors is that DI garnets contain on average more V than X garnets. While this is apparently true, given the mostly uncorrected DI dataset, the authors' analysis fails to mention that the DI dataset and the X dataset bracket nearly identical spectrums of V contents, with V contents in the X garnets extending to even higher values than the DI garnets (eg Supplementary Figure 1). This seems important to emphasize in the text.
 - b. **General correlations between model parameters and garnet crystal chemical effects.** The authors attribute increased V in DI garnets to coupled uptake with Na, but both the tables and the figures suggest there is no strong correlation between Na₂O and V in garnet. While the relationship between Na₂O and V in garnet may be statistically significant as shown by the authors p-value tests, there is no strong correlation between V and Na in garnet with the exception of suite 6 ($R^2=0.7$; Table 2). This conclusion is overstated given the authors' modeling results.

This lack of correlation between parameters extends to the general modeling in Table 2, which models chemical trends in the authors data set for specific cratons. Table 2 contains 46 R^2 values for modeled relationships in the authors dataset. Twenty of the modeled parameters in show no or very weak correlation ($R^2<0.5$) despite the existence of statistically significant relationships (p-value<0.05) between some model parameters. Another 12 are moderately correlated ($R^2=0.5-0.7$) and only 10 are strongly correlated ($R^2>0.7$). There are few strong correlations present in the dataset, with the exception of Ti vs V in garnet—see point #1—and for garnet Na₂O vs. T. Given the lack of coherent trends even within single cratons, it seems unlikely that any single process or substitution

mechanism can be identified as the driving force for V substitution in garnet, cpx or bulk eclogites globally. I recommend the authors use more circumspect language to reflect this, particularly in the abstract of the manuscript.

- c. **Vanadium valence.** The authors infer that DI garnets contain substantial V^{4+} , a surprising conclusion considering the generally low fO_2 s required for diamond growth. This conclusion appears to be drawn from a comparison of eclogite fO_2 s and V valences measured in silicate glass standards. Vanadium valences in silicate glass standards are plotted as a function of fO_2 as a second y-axis in the authors' figures throughout the paper. However, V valences measured in silicate glasses equilibrated at high T (1300-1400 °C) and 1 atm are not representative of V valences in eclogitic minerals equilibrated at much lower T and higher P. Vanadium valence is expected to change as a function of temperature and pressure (similar to effects known for Fe), but there is no existing literature data that could be used to infer the valence of V in eclogitic minerals. The extrapolation of the silicate glass data to eclogitic minerals is unwarranted.
3. **Cpx data.** The authors conclude that the higher V contents of DI garnets compared to X garnets is due to increased Na uptake. If this is true, this trend should theoretically extend to cpx—V should be higher in DI cpx compared to X cpx since the jadeite content of cpx is strongly dependent on pressure and to a lesser degree, temperature. V in both garnet and cpx is inferred to occupy the octahedral site, substituting for Al. However, the range of V in both the X and DI cpx overlap completely and there is no correlation of V in cpx with reported jadeite content. Cpx Na_2O content is also not reported in the Supplement. It is not clear why the uptake of V in DI garnet would be subject to crystal chemical effects but DI cpx would not be, and this is not explained by the authors. This should be covered in the main text of the manuscript prior to publication.
4. **Modeling.** All model inputs and outputs must be reported in the Supplemental Material so that readers may replicate the authors results.

Reviewer #2 (Remarks to the Author):

In this manuscript, Aulbach and Stachel present literature data on the chemical composition of cpx and garnet from cratonic eclogites compared with eclogitic garnet and cpx trapped in diamonds essentially from the same rocks. They propose V as element to reveal variations in P-T-X-fo₂ of both rock and diamonds and they conclude that diamonds preserve pristine conditions at the time of metamorphism of metabasalts, while eclogitic rocks experienced multiple events that might have affected their chemistry in terms of trace and major elements.

The main result of this paper is that the reconstructed V concentrations of modeled eclogite rocks from the garnet/cpx inclusions in diamonds and from the analyzed host rock has not changed much implying an efficiently buffered system to O₂ during subduction over time. This might have resulted in eclogites being the most important carrier of diamonds compared to peridotites.

The work is significant in the eclogite and diamond community whose the authors are among the most prominent leaders. All existing useful literature is properly referenced that make the reconstruction model of the bulk chemistry of eclogites impressively robust. The authors took into account all possible effects to describe the variation of V both in cpx and grt even in presence of accessory minerals and fluid interaction.

Only few comments are regarding 1) how can one be sure that inclusions in diamonds have been equilibrated for the same duration over which the rock matrix experienced chemical and temperature variations? After all, these are non-touching inclusions; 2) are the proposed models of V partitioning also applicable in the case of hydrous carbonated fluids? The authors seems to cite experimental studies on V partitioning in the case of silicate (e.g. tonalitic) melts and not carbonatitic melts; 3) how changes in the modal composition of eclogites affect the fo₂? What is the effect of SiO₂ and Al₂O₃ saturation (with appearance of coesite and kyanite, respectively) on V partitioning?

To my knowledge NatureComm requires a more fluent text that can be understandable to a broad scientific community. Some of the technical discussion should be integrated to the Supplemental Materials and provide more emphasis to the implications of this study in terms of volatile transport and mantle oxidation.

Best wishes

Reviewer #3 (Remarks to the Author):

Aulbach and Stachel (2021) uses the extensive database of published major and trace element data for eclogites and diamond inclusions in kimberlites to make some very interesting and relevant conclusions about the oxygen fugacity of subducted oceanic rocks and the redox effects of diamond formation in eclogite. The paper overall reflects the authors' immense knowledge on the topic of eclogite geochemistry and their command of the literature and should be published in Nature Communications once some minor issues have been fixed. In-line comments, mainly grammatical in nature, are also added at the end of the review.

Firstly, and perhaps more importantly, I feel like the authors undersell the big point of the paper, not even mentioning it in the abstract. The fact that metasomatism related to diamond formation does not appear to alter the fO_2 of eclogites is a big point and should be emphasized in the abstract, which presently is a bit overly specific and confused. I would also suggest shortening the title of the paper for more impact to simply: "Evidence for redox-neutral eclogitic diamond formation" so that the reader immediately knows the great significance of the main conclusion. Secondly, I am very interested in the authors conclusion in Lines 262-263 that xenoliths are much more likely to be gabbroic in nature than diamond inclusions. I feel like this is mentioned in a throw-away line and should at least have its own paragraph discussing this conclusion. Is this conclusion statistically robust or does it reflect the small number of available DI samples? Why would diamonds not enclose gabbroic minerals? Is the metasomatism that formed diamonds only effective in basaltic eclogites? Or does the metasomatism producing diamonds remove geochemical indicators of cumulate protoliths? Are gabbroic minerals somehow resistant to being included? This definitely needs more discussion, and I was left wondering about it. Finally, it would help the reader if the authors would provide brief explanations of how the conclusions of some of the cited papers were reached. For instance, it is cited that metasomatism can be oxidative or reductive in the mantle, but I was left wondering how the fO_2 was calculated. Also how are the eclogites dated? By kimberlite ages or by somehow directly dating eclogites themselves? Are individual eclogites shown to differ in age within a single suite? I understand that the authors are themselves experts in eclogite geochemistry, and that this information is in the cited papers, but a tad more background throughout would be helpful to the reader.

Overall interesting stuff, and I hope you two have plans for more LA-ICP-MS studies of diamond inclusions in the future as I'd love to see more data and conclusions for this story!

-R. Willie Nicklas
Scripps Isotope Geochemistry Laboratory
University of California, San Diego

In-Line Comments:

- Line 29: use "and" not ";"
- Line 33-36: this sentence is confusing, is the dichotomy caused by crystal chemistry, temperature, or fO_2 ? Or all three?
- Line 47: change to "evolution of these parameters", "its" is ambiguous
- Line 55: you define oxygen fugacity here, but mention it earlier in the paragraph, perhaps start the introduction by defining oxygen fugacity?
- Line 59: you don't need to define an acronym twice
- Line 71: "in" to "within"
- Line 80: "these" findings
- Lines 92-93: define for the reader that "Kaapvaal" and "Slave" are Archean cratons
- Line 95: in "part"
- Line 102: "Geochronological" instead of "Dating"
- Line 105: "Have been" detected
- Line 115: "each" eclogite reservoir
- Line 128: "depends" on fO_2
- Line 147: how is rutile mode estimated for a monomineralic DI? Give us a one sentence explanation of how this calculation is performed.
- Line 158: rutile solubility "effects"
- Line 174: do we think garnet is only taking up V+4? Care to speculate on whether it can dissolve V+3 and what site we might expect it to take?
- Line 186: Wait I'm confused, are we considering natural or experimental partition coefficients? Be clear here. Also how is fO_2 estimated in these natural samples, could some of the scatter come from poorly constrained fO_2 estimates?

- Line 203: This feels like a re-hash of the previous section, especially for such a short paper let's not repeat ourselves too much. Either put the fO₂ section before the crystal chemistry one or include the substitution equations in the crystal chemistry section where they make more sense.
- Line 222: a trivalent "cation" plus a divalent cation
- Line 244: are we assuming we are melting primitive mantle at all points in Earth's history of DMM? Shouldn't the mantle become more depleted with time? Either explain what mantle V abundance we are assuming or do a calculation to justify that it shouldn't matter for the conclusions of this paper.
- Line 248: change "related to" to "for"
- Line 268: "basaltic" melt protoliths
- Line 285: Northern Slave and Kaapvaal "cratons"
- Line 290: "the" incompatibility and "the" compatibility
- Line 310: how is fO₂ estimated in the 2017 paper you reference here for oxidizing and reducing metasomatism? Not using V partitioning correct? As that would make it circular logic. Indicate how you determined fO₂ in those Greenlandic eclogites.
- Line 346: I may be mistaken, but doesn't carbonate melt not transport HFSE such as Nb, Zr and Hf? Why would HFSE enrichment make you think it was carbonate metasomatism?
- Line 377: wouldn't this only be true if melting of eclogite was happening at the same fO₂ as mantle melting that formed its protolith?
- Line 384: Does "paired" mean in the same inclusion or as separate inclusions within the same diamond?
- Line 390: how many of the xenoliths contained the same diamonds used for the DI data? or are they all separate entities? Where the diamonds included in the eclogites or separate entities within the kimberlites?
- Line 403: specify that Sc is homovalent and trivalent, i.e. analogous only to trivalent V
- Line 410: ppm or wt.%? I'm confused
- Line 430: "The" two data-sets "used in this study"
- Line 445: for both "types of sample"
- Line 468: within "uncertainties"

Point-by-point response to the reviewers' comments

Reviewer comments are reproduced verbatim below in black font, our replies in blue font. Bold blue font points to the sections in the manuscript where corresponding changes were effected.

Reviewer #1 (Remarks to the Author):

See attached PDF of review.

In short:

I recommend that this manuscript undergo substantial revision, and because the authors conclusions may change, that any revised versions of this manuscript undergo peer review.

From the attached pdf:

Aulbach and Stachel collate and filter a global dataset to compare concentrations of V in garnet and cpx in eclogitic xenoliths (X) with inclusions of these minerals in diamonds (DI) from the same locality. This comparison is useful for examining the effect of metasomatism and melt extraction on the V composition of eclogitic minerals because diamond inclusions are shielded from these processes post-encapsulation. The authors find the mean concentrations of V in DI garnets are greater than in X garnets from the same locality, with the increase facilitated by coupled substitution of V with Na in DI garnets. The authors reconstruct the bulk rock compositions from cpx and garnet data to assess the impact of low-pressure differentiation (i.e., melting at MOR) and partial melt extraction and metasomatism at high pressure on the V contents of eclogites. A key conclusion of this manuscript is that DI garnets appear to contain oxidized species of V (V⁴⁺) despite their inferred low oxygen fugacities.

The general premise of this manuscript is sound and the dataset comparing DI to X eclogitic minerals will be of interest to many scientists. However, the authors' conclusions are overstated given the broad scatter and lack of coherent trends in their dataset. Some conclusions, such as the presence of V⁴⁺ in DI garnets, are not supported by this data presented in this study or by literature data.

We very much thank the reviewer for pointing out the problems with our inference that V⁴⁺ might be present in garnet. We now agree that this is unlikely and have changed our interpretation accordingly.

Importantly, the DI dataset has not been uniformly corrected for EPMA Ti-V overlap and as such cannot be used for comparison, modeling and interpretation.

We appreciate the reviewer's concern, but the DI dataset used in this study has been uniformly corrected. The diamond research community has been aware for decades of the EPMA Ti-V overlap problem. Consequently, EPMA-derived V concentrations have, in fact, been routinely corrected.

I recommend that this manuscript undergo substantial revision, and because the authors conclusions may change, that any revised versions of this manuscript undergo peer review.

Specific major comments follow.

1. Ti-V correction of the DI data set. The diamond inclusion dataset, mostly collected by electron probe microanalysis, has not been uniformly corrected for overlap of the TiK β and VK α peaks. The impact of this overlap is that TiK β counts will be erroneously attributed to VK α such that the calculated concentrations of V are higher than ‘true’ values. The effect scales with Ti content—as Ti increases, the overlap worsens and calculated V contents increase. The degree of overlap is also highly dependent on what crystals are used in the EPMA analytical routine because the TiK β peak is closer to the VK α peak on LPET crystals than on LLIF crystals. The overlap is about 10x ‘worse’ with LPET compared to LLIF, with about 0.6% of counts on pure Ti reference materials attributed to V with analysis using LLIF crystals vs about 6% of counts with LPET crystals. Note that it is typically more common to analyze these elements on LPET crystals because they yield higher count rates compared to LLIF. Both elements must be analyzed on the same crystal to apply the overlap correction. Analyzing V on LLIF and Ti on LPET, for example, does not avoid or eliminate the overlap issue.

We fully agree with the reviewer. The community standard has been to use LIFH or LIFL for V analyses for exactly the reasons outlined by the reviewer, and to apply a correction for Ti overlap. We do not concur that both elements must be analysed on the same crystal for accurate correction; this view is confirmed by Dr. A. Locock (pers. comm. 28 Sep 21), a renowned mineralogist and electron microprobe specialist at the University of Alberta.

A key conclusion of the authors is that DI garnets contain more V than X garnets, which they attribute to the presence of increased Na in the DI garnets to charge balance V. However, the DI garnets contain on average more Ti than X garnets, raising the possibility that the increase in V is simply due to uncorrected Ti-V overlap in EPMA analyses. Nearly all DI data in this study are from EPMA measurements, but only a small subset of these data are reported as being corrected for Ti-V overlap (lines 413-417) [note analyses of X garnets using LAM-ICP-MS are not subject to this overlap].

All EPMA-derived data used in this study were corrected. **We now explicitly state this in the Methods section under “Assessment of analytical methods...”, and we have added information to column D in Supplementary Table 1 where we specify the type of correction applied.**

The authors’ assumption that TiO₂ contents <1.76 wt% are unproblematic is also not acceptable for the purposes of this study.

The sentence the reviewer refers to is indeed ambiguous. What we meant is that Wang et al. (2019, cited ref. 35) consider TiO₂ contents up to 1.76 wt.% unproblematic in terms of applying an accurate correction. **We have amended the sentence in the Methods section under “Assessment of analytical methods...” to clarify this.**

For example: assuming for simplicity’s sake a 1:1 relationship between counts and concentration, analysis of a garnet containing 0.68 wt% TiO₂ (e.g., DI sample 245 from Kaapvaal) would result in 277 ppm of Ti that would be erroneously attributed to V without overlap correction. This is more than enough ‘vanadium’ to significantly impact the authors results and interpretations.

The reviewer’s example is illustrative, but very pessimistic. The senior author’s (Stachel) experience is that ~60-70 ppm V per wt.% TiO₂ result pre-overlap correction from using LIF-H on older-generation instruments (late 90ies, early 2000s). To further investigate this matter, Aulbach, with the help of PD Dr. Dominik Hezel, carried out an experiment measuring V (using a pure V metal standard and LIFL crystal

for quantification) in synthetic rutile and in clinopyroxene \pm garnet in three eclogite samples by EPMA, for which V concentrations had been independently determined by LAM ICPMS, both at Goethe-University Frankfurt. Results show that the average apparent V attributable to the overlap is 26.7 ± 0.7 ppm per wt.% TiO₂, corresponding to 4% of the total V determined in clinopyroxene and 11% in garnet; corrected EPMA-derived results agree within the uncertainty with LAM-ICPMS-derived results, and are reported in **new Table S7, shown in Fig. S4 (formerly S5) and described in paragraph four of the Methods section “Assessment of analytical methods...”**.

The potential impact of Ti-V overlap can be seen in the DI data collated in the Supplement. For instance, the dataset of Phillips et al 2004 clearly shows a linear increase in both V in garnet and cpx as TiO₂ content of both minerals increases, as would be expected if a Ti-V correction is not deployed.

We contacted Prof. Dave Philipps at University of Melbourne (dphillip@unimelb.edu.au). We cite from his email dated 28 Sep 2021: “I do recall making interference corrections including the Ti interference on V, as this is a well-known issue”. We are therefore confident that the mentioned correlation between V and TiO₂ reflects crystal chemical effects.

Prof. Fanus Viljoen at University of Johannesburg (fanusv@uj.ac.za) replied via email on 19 Oct 2021 that “Samantha Perritt [second author on the paper] would have used a microprobe routine which accounts for Ti overlap on V, for the analyses conducted at UJ”.

The senior author (Stachel) was involved as author or co-author in the production of the remaining EPMA-derived DI datasets used in this study, which were collected at University of Alberta and corrected either off-line using a pure TiO₂ standard, or on-line using the Donovan “Probe for EPMA” software installed on the instrument.

In sum: vanadium concentrations from EPMA are meaningless without application of a Ti-V overlap. This manuscript cannot be published containing V data from the EPMA that have not be subject to a Ti-V overlap.

We completely agree and can say with confidence that all data used in this study have been corrected.

2. Overstating of conclusions. Many of the conclusions in this manuscript are overstated and do not adequately reflect the breadth and scatter apparent in the authors dataset.

Scatter is always a problem in mantle-derived xenoliths (as opposed to melts). In eclogite xenoliths and inclusions in diamond, this is due to the multiplicity of protoliths and processes that the samples experienced. This leads to a wide range of major and trace element concentrations in bulk rocks and minerals. These various origins and processes can be recognised using appropriate elemental or isotopic proxies (Aulbach and Jacob 2016, Aulbach et al. 2020, cited refs. 57,21), as also done in this study. It is inevitable that these processes have in part opposing effects on the contents of V and other elements. In terms of V incorporation into minerals, crystal-chemistry and pressure-temperature-oxygen fugacity further complicate the picture, as we discuss. In this light, the scatter is an inherent property of this type of samples. We would rather try and make sense of the systematics than dismissing these samples due to their complexity. We suggest that crystal-chemical and other controls are recognisable in the fact that statistically significant signatures can still be isolated. **We have added a small paragraph (directly under “3. Crystal-chemical and petrologic controls...”)** to the main text reflecting the aforesaid, including

emphasising the weakness of some correlations. This puts the reader in a position where they can decide how much faith they wish to place into the conclusions drawn based on these relationships.

Other conclusions are simply not supported by the literature data. Specific examples:

a. V in garnet comparison. Setting aside the possible impact of Ti-V overlap on the DI garnet data, an important conclusion of the authors is that DI garnets contain on average more V than X garnets. While this is apparently true, given the mostly uncorrected DI dataset, the authors' analysis fails to mention that the DI dataset and the X dataset bracket nearly identical spectrums of V contents, with V contents in the X garnets extending to even higher values than the DI garnets (eg Supplementary Figure 1). This seems important to emphasize in the text.

Please see reply under (1.) above. Since the DI dataset is corrected, the higher V contents in DI garnet are not apparent, but geologically interpretable. It is correct that the xenolith garnet V contents range to even higher values than DI garnet V; this is particularly so for the suite from the northern Slave craton, which also shows anomalous (relative to the other cpx-garnet pairs) distribution of V between cpx and garnet (Fig. S1a,c), as is discussed **in the 3rd paragraph of section 4.2 "Counterintuitive effects" where we now emphasise that these concentrations are higher than in any DI.** Apart from that, Fig. S1b also shows that V in DI garnet does not extend to the low values observed in their xenolithic counterparts, which is unsurprising given their significantly different averages in Fig. 1a.

b. General correlations between model parameters and garnet crystal chemical effects. The authors attribute increased V in DI garnets to coupled uptake with Na, but both the tables and the figures suggest there is no strong correlation between Na₂O and V in garnet. While the relationship between Na₂O and V in garnet may be statistically significant as shown by the authors p-value tests, there is no strong correlation between V and Na in garnet with the exception of suite 6 (R²=0.7; Table 2). This conclusion is overstated given the authors' modeling results.

This lack of correlation between parameters extends to the general modeling in Table 2, which models chemical trends in the authors data set for specific cratons. Table 2 contains 46 R² values for modeled relationships in the authors dataset. Twenty of the modeled parameters in show no or very weak correlation (R²<0.5) despite the existence of statistically significant relationships (p-value<0.05) between some model parameters. Another 12 are moderately correlated (R²=0.5-0.7) and only 10 are strongly correlated (R²>0.7). There are few strong correlations present in the dataset, with the exception of Ti vs V in garnet—see point #1—and for garnet Na₂O vs. T. Given the lack of coherent trends even within single cratons, it seems unlikely that any single process or substitution mechanism can be identified as the driving force for V substitution in garnet, cpx or bulk eclogites globally. I recommend the authors use more circumspect language to reflect this, particularly in the abstract of the manuscript.

We now refrain from making specific predictions about which coupled substitutions V may be involved in, and have modified the abstract and main text (third paragraph in "3.1 Temperature and crystal-chemical effects") to reflect this. We have also deleted original Fig. S2 illustrating suggested substitution mechanisms, and removed corresponding columns from Table 2.

c. Vanadium valence. The authors infer that DI garnets contain substantial V⁴⁺, a surprising conclusion considering the generally low fO₂s required for diamond growth. This conclusion appears to be drawn

from a comparison of eclogite fO_2 s and V valences measured in silicate glass standards. Vanadium valences in silicate glass standards are plotted as a function of fO_2 as a second y-axis in the authors' figures throughout the paper. However, V valences measured in silicate glasses equilibrated at high T (1300-1400 °C) and 1 atm are not representative of V valences in eclogitic minerals equilibrated at much lower T and higher P.

Vanadium valence is expected to change as a function of temperature and pressure (similar to effects known for Fe), but there is no existing literature data that could be used to infer the valence of V in eclogitic minerals. The extrapolation of the silicate glass data to eclogitic minerals is unwarranted.

We thank the reviewer for pointing out issues of translating V speciation in glasses to high-pressure garnet, and accept that there is no case for tetravalent V. We have **modified Figures 2, 4 and discussion accordingly. This has also prompted a new interpretation of the, albeit weak, correlation between fO_2 and V abundance (section 4.2 "Counterintuitive effects"), and to addition of a new panel c to Figure 4.**

On further discussion with Dr. Locock, we were made aware of a paper where V speciation was measured by XANES in an Fe³⁺ rich schorlomite megacryst from an alkaline complex, i.e., representing garnet at fairly oxidised conditions (Locock et al. 1995, now cited ref. 33). The authors find exclusively trivalent V, in octahedral coordination. On the other hand, V abundance in garnet and bulk rocks is not independent of fO_2 , despite the scatter, and this must reflect some underlying redox control. This is, after all, the rationale in using bulk V in mantle-derived melts as a proxy for fO_2 during the melting process, i.e. at high pressure in the mantle source (e.g. Lee et al. 2005 and Mallmann and O'Neill 2009, cited ref. 12,13). Righter et al. (cited ref. 29) find decreasing garnet-melt distribution coefficients but a rather constant V valence state of $\sim +2.5$ (i.e. $\sim 50\% V^{2+}$ and $50\% V^{3+}$) over an extended fO_2 range between FMQ-4.5 to FMQ-1.8 in high-pressure experiments and natural garnets, suggesting that garnet crystal-chemistry exerts a stronger control on V valence than fO_2 . **We have added this information to the second paragraph of section "3.2 Oxygen fugacity effects...", without making specific predictions for V speciation.**

3. Cpx data. The authors conclude that the higher V contents of DI garnets compared to X garnets is due to increased Na uptake. If this is true, this trend should theoretically extend to cpx—V should be higher in DI cpx compared to X cpx since the jadeite content of cpx is strongly dependent on pressure and to a lesser degree, temperature. V in both garnet and cpx is inferred to occupy the octahedral site, substituting for Al. However, the range of V in both the X and DI cpx overlap completely and there is no correlation of V in cpx with reported jadeite content. Cpx Na₂O content is also not reported in the Supplement. It is not clear why the uptake of V in DI garnet would be subject to crystal chemical effects but DI cpx would not be, and this is not explained by the authors. This should be covered in the main text of the manuscript prior to publication.

We did do not say, or imply, that V uptake in cpx is not subject to crystal chemical effects; **we now specifically mention this in paragraph 1 under "3.1 Temperature and crystal-chemical effects..."**. We note here that Na₂O in clinopyroxene can be regarded as an infinite reservoir for the substitution of trace elements, whereas in garnet it is of a similar order of magnitude as V (mostly <1000 ppm Na), and may therefore have a limiting effect on substitutions involving Na. As we report jadeite content in clinopyroxene, which is strongly correlated with clinopyroxene Na₂O content ($r^2 = 0.95$ for $n = 380$), we feel that addition of Na₂O to the database will cause redundancy.

4. Modeling. All model inputs and outputs must be reported in the Supplemental Material so that readers may replicate the authors results.

Yes, in addition to the details already given in the Methods section, **we now include new Supplementary Tables 2-6, with explicit information on how the numbers were obtained or where they were taken from.**

Reviewer #2 (Remarks to the Author): 7

In this manuscript, Aulbach and Stachel present literature data on the chemical composition of cpx and garnet from cratonic eclogites compared with eclogitic garnet and cpx trapped in diamonds essentially from the same rocks. They propose V as element to reveal variations in P-T-X-fo₂ of both rock and diamonds and they conclude that diamonds preserve pristine conditions at the time of metamorphism of metabasalts, while eclogitic rocks experienced multiple events that might have affected their chemistry in terms of trace and major elements.

The main result of this paper is that the reconstructed V concentrations of modeled eclogite rocks from the garnet/cpx inclusions in diamonds and from the analyzed host rock has not changed much implying an efficiently buffered system to O₂ during subduction over time. This might have resulted in eclogites being the most important carrier of diamonds compared to peridotites.

The work is significant in the eclogite and diamond community whose the authors are among the most prominent leaders. All existing useful literature is properly referenced that make the reconstruction model of the bulk chemistry of eclogites impressively robust. The authors took into account all possible effects to describe the variation of V both in cpx and grt even in presence of accessory minerals and fluid interaction.

Only few comments are regarding

1) how can one be sure that inclusions in diamonds have been equilibrated for the same duration over which the rock matrix experienced chemical and temperature variations? After all, these are non-touching inclusions;

Indeed, isolated inclusions stopped equilibrating at the time of encapsulation, freezing in concentrations that would reflect temperatures and garnet-clinopyroxene equilibria at the time of diamond formation – that's part of their utility. If they were touching and able to re-equilibrate, we would not be able to see e.g. the temperature effect on V distribution as well as we do. **We added a sentence to this effect in paragraph 1 of the Introduction.**

2) are the proposed models of V partitioning also applicable in the case of hydrous carbonated fluids? The authors seems to cite experimental studies on V partitioning in the case of silicate (e.g. tonalitic) melts and not carbonatitic melts;

That's a good question. The effect of volatiles on trace-element partitioning is known (e.g. Blundy and Dalton 2000 Contrib Mineral Petrol), but we do not currently have the experimental database to assess the effect fO₂ on V partitioning in carbonated systems. Wang et al. 2019 (cited ref. 35) find that the

partitioning of V, Sc and Ti is independent of H₂O in the melt. Both partial melting at spreading ridges – where the protoliths to the eclogites formed – and partial melting of metabasalt during subduction are silicate rather than carbonated systems. The latter might become important during subsequent metasomatic processes. We had written in original **paragraph 2 of the Discussion** “...V concentration in metasomatic clinopyroxene is modelled assuming that the fO₂-dependent clinopyroxene-melt partitioning (parameterised by ref. 35) also applies for carbonated silicate melts, such as kimberlites.” This has been moved to the last paragraph in **Methods**. To account for the reviewer’s comment, **we added “for which redox-dependent V partitioning data are not yet available”**.

3) how changes in the modal composition of eclogites affect the fo₂?

Aulbach et al. 2017 (Earth Planet Sci Lett) inferred that an increasing garnet mode with depth, occurring in metabasalts as demonstrated in subsolidus experiments by Knapp et al. 2015 (J Geophys Res), causes a dilution of Fe³⁺ and is responsible for the weak decrease in fO₂ with depth observed in eclogite xenoliths from the Lace kimberlite. Stagno et al. 2015 (cited ref. 70), who formulated the eclogite oxybarometer, calculated a weak decrease in fO₂ with depth, but did not tie this to changing modal compositions. This is a very interesting topic, but we feel it is too early to discuss it with any confidence.

What is the effect of SiO₂ and Al₂O₃ saturation (with appearance of coesite and kyanite, respectively) on V partitioning?

Most eclogite xenoliths from the literature are essentially bimineralic rocks. In a manuscript by Stachel, Aulbach and Harris (accepted for publication in Reviews in Mineralogy and Geochemistry; new ref. 34) (*kindly note V systematics are NOT discussed in that paper*), we find that “Of 2451 xenoliths considered for comparison of mineralogy, commonly reported primary accessory and minor minerals include ... kyanite (n=67) ... and coesite/quartz (n=11).” V might be slightly compatible in kyanite (Brey et al. 2015 CMP), but this mineral is generally trace-element poor (e.g. kyanite in a Kaapvaal eclogite xenolith contains 28 ppm V; Schmickler et al. 2004 Lithos). V might be also slightly compatible in coesite (Yan et al. 2021 Sci China Earth Sci). However, the rarity of these minerals, their low abundance when present, and the only mildly compatible nature of V, if that, suggest that they cannot affect the data structure in the present study.

To my knowledge NatureComm requires a more fluent text that can be understandable to a broad scientific community. Some of the technical discussion should be integrated to the Supplemental Materials and provide more emphasis to the implications of this study in terms of volatile transport and mantle oxidation.

We infer that this refers to the many modelling details. Since relevant literature that is relegated to the supplement will not garner citations as per NatComms policy, **we opted to move some of the modelling detail to the Methods**, where those interested can still read up on it, and we deleted some text in the methods repeating interpretations that are presented in the discussion to minimise duplication.

We now provide more discussion on the implications, including volatile transport and mantle oxidation in the last **section (4.3 “Redox-neutral diamond formation...”)**.

We also renamed the results (now “Crystal-chemical and petrologic controls on V abundances in eclogite minerals” and discussion (now Implications) section and provided subheaders for the latter to make it more obvious for the reader what the main thrust of these various sections is.

Reviewer #3 (Remarks to the Author):

Aulbach and Stachel (2021) uses the extensive database of published major and trace element data for eclogites and diamond inclusions in kimberlites to make some very interesting and relevant conclusions about the oxygen fugacity of subducted oceanic rocks and the redox effects of diamond formation in eclogite. The paper overall reflects the authors’ immense knowledge on the topic of eclogite geochemistry and their command of the literature and should be published in Nature Communications once some minor issues have been fixed. In-line comments, mainly grammatical in nature, are also added at the end of the review.

Firstly, and perhaps more importantly, I feel like the authors undersell the big point of the paper, not even mentioning it in the abstract. The fact that metasomatism related to diamond formation does not appear to alter the fO_2 of eclogites is a big point and should be emphasized in the abstract, which presently is a bit overly specific and confused.

Agreed, **we have changed the title, and now emphasise this point at the end of the abstract where we also deleted some less compelling observation.**

I would also suggest shortening the title of the paper for more impact to simply: “Evidence for redox-neutral eclogitic diamond formation” so that the reader immediately knows the great significance of the main conclusion.

We adapted the title as suggested by the reviewer but also refer to the redox-buffered nature of subducted oceanic crust, a point highlighted as particularly noteworthy by reviewer 2.

Secondly, I am very interested in the authors conclusion in Lines 262-263 that xenoliths are much more likely to be gabbroic in nature than diamond inclusions. I feel like this is mentioned in a throw-away line and should at least have its own paragraph discussing this conclusion.

Why would diamonds not enclose gabbroic minerals?

Is the metasomatism that formed diamonds only effective in basaltic eclogites?

Are gabbroic minerals somehow resistant to being included? This definitely needs more discussion, and I was left wondering about it.

These are good questions. The observation extends even to diamondiferous eclogites, which also show gabbroic signatures more frequently than inclusions (Stachel et al. RIMG, accepted). We did not go into this because we can only speculate, but we have now **added a tentative explanation in paragraph 2 of new section 3.3.**

Or does the metasomatism producing diamonds remove geochemical indicators of cumulate protoliths?

Is this conclusion statistically robust or does it reflect the small number of available DI samples?

Given that Al_2O_3/FeO , which we suggest as gauges for gabbroic vs. non-gabbroic protoliths, is significantly different for xenolithic and DI clinopyroxene and garnet (Table 1), this conclusion seems statistically robust – **a point now made in the text.**

Finally, it would help the reader if the authors would provide brief explanations of how the conclusions of some of the cited papers were reached. For instance, it is cited that metasomatism can be oxidative or reductive in the mantle, but I was left wondering how the fO_2 was calculated.

We have added to section 4.2 “Counterintuitive effects...” that oxidising and reducing metasomatism were recognised based on oxybarometry applied to peridotite xenoliths.

Also how are the eclogites dated? By kimberlite ages or by somehow directly dating eclogites themselves? Are individual eclogites shown to differ in age within a single suite?

We now provide a brief explanation and reference to a recent paper on eclogite dating in paragraph 2 of section 2 “Background”.

I understand that the authors are themselves experts in eclogite geochemistry, and that this information is in the cited papers, but a tad more background throughout would be helpful to the reader.

We much appreciate the reviewer’s curiosity about eclogites and mantle metasomatism. There is not enough space to accommodate all aspects of eclogite petrogenesis, for which references mostly to review papers are provided. As part of the revisions, we now specifically address the above questions.

Overall interesting stuff, and I hope you two have plans for more LA-ICP-MS studies of diamond inclusions in the future as I’d love to see more data and conclusions for this story!

-R. Willie Nicklas
Scripps Isotope Geochemistry Laboratory
University of California, San Diego

In-Line Comments:

-Line 29: use “and” not “;”

Done

-Line 33-36: this sentence is confusing, is the dichotomy caused by crystal chemistry, temperature, or fO_2 ? Or all three?

Sentence changed in response to reviewer 1

-Line 47: change to “evolution of these parameters”, “its” is ambiguous

Done

-Line 55: you define oxygen fugacity here, but mention it earlier in the paragraph, perhaps start the introduction by defining oxygen fugacity?

Done

-Line 59: you don't need to define an acronym twice

Fixed

-Line 71: "in" to "within"

Done

-Line 80: "these" findings

Done

-Lines 92-93: define for the reader that "Kaalvaal" and "Slave" are Archean cratons

Done

-Line 95: in "part"

Done

-Line 102: "Geochronological" instead of "Dating"

Done

-Line 105: "Have been" detected

Done

-Line 115: "each" eclogite reservoir

Done

-Line 128: "depends" on fO_2

Done

-Line 147: how is rutile mode estimated for a monomineralic DI? Give us a one sentence explanation of how this calculation is performed.

This is done in the Methods section under "Bulk rock reconstruction and effect of rutile", and in footnote 2 of Table 1

-Line 158: rutile solubility "effects"

Done

-Line 174: do we think garnet is only taking up V+4? Care to speculate on whether it can dissolve V+3 and what site we might expect it to take?

At the behest of Reviewer 1, we now refrain from making specific predictions as to V speciation or crystal site

-Line 186: Wait I'm confused, are we considering natural or experimental partition coefficients? Be clear here. Also how is fO_2 estimated in these natural samples, could some of the scatter come from poorly constrained fO_2 estimates?

We now clarify that both experimental and natural D values are considered (2nd sentence, section 3.2).

-Line 203: This feels like a re-hash of the previous section, especially for such a short paper let's not repeat ourselves too much. Either put the fO_2 section before the crystal chemistry one or include the

substitution equations in the crystal chemistry section where they make more sense.

This paragraph has been deleted to comply with comments from Reviewer 1

-Line 222: a trivalent “cation” plus a divalent cation

This paragraph has been deleted to comply with comments from Reviewer 1

-Line 244: are we assuming we are melting primitive mantle at all points in Earth’s history of DMM? Shouldn’t the mantle become more depleted with time? Either explain what mantle V abundance we are assuming or do a calculation to justify that it shouldn’t matter for the conclusions of this paper.

Added to the main text that a DM starting composition with 73 ppm V is used by Wang et al. (ref. 35) in their spreadsheet and adopted here. To keep technicalities short as per Reviewer 2, we added to the Methods section: Wang et al. (2019)³⁵ employ a Depleted Mantle starting concentration of 73 ppm, which is adopted, consistent with findings that the mantle source of the protoliths to eclogite xenoliths was depleted (if heterogeneous) by ca. 3.0 Ga ago, the age of the oldest suite.

Besides, the effect of a Primitive vs. Depleted Mantle starting composition and mineralogy was explored by Aulbach and Stagno 2016 (Geology; their Supplemental Table 4) who find that the differences between the sources are too small to explain differences in V/Sc across the Archaean-Proterozoic boundary. At any rate, a change in CO would only effect a shift in the isopleths and not affect the relative position of Palaeoproterozoic vs. Archaean eclogite suites shown in Fig. 5.

Line 248: change “related to” to “for”

Done

-Line 268: “basaltic” melt protoliths

Done (basaltic to picritic)

-Line 285: Northern Slave and Kaapvaal “cratons”

Done

-Line 290: “the” incompatibility and “the” compatibility

Done

-Line 310: how is fO₂ estimated in the 2017 paper you reference here for oxidizing and reducing metasomatism? Not using V partitioning correct? As that would make it circular logic. Indicate how you determined fO₂ in those Greenlandic eclogites.

It was conventional garnet-based oxybarometry – this info added to the text

-Line 346: I may be mistaken, but doesn’t carbonate melt not transport HFSE such as Nb, Zr and Hf? Why would HFSE enrichment make you think it was carbonate metasomatism?

Yes, pure carbonate melt (carbonatite) at mantle depth is not thought to add HFSE. In contrast, kimberlite is a carbonated (CO₂-rich) melt that can transport HFSE. Replaced “carbonated” with CO₂-rich to avoid confusion yet remain within the cited authors’ interpretations

-Line 377: wouldn’t this only be true if melting of eclogite was happening at the same fO₂ as mantle melting that formed its protolith?

[Note this originally last discussion paragraph is now the first, i.e. 4.1]

Yes, good point. The important thing is that V switches from behaving incompatibly at FMQ to incompatibly at ~FMQ-1 and below. Since most eclogites have low fO_2 , their V would have increased, if anything, during high-pressure partial melt loss. We added corresponding information to the Methods, now additionally model FMQ-1 and FMQ-4 in new Supplementary Table 5, show the two extremes (FMQ and FMQ-4) in Fig. 4a and Fig. 5 and discuss them in the third paragraph of section 3.4, and again in section 4.1.

-Line 384: Does “paired” mean in the same inclusion or as separate inclusions within the same diamond?
Clarified that they are separate inclusions in the same diamond, which, rarely, may be touching

-Line 390: how many of the xenoliths contained the same diamonds used for the DI data? or are they all separate entities? Where the diamonds included in the eclogites or separate entities within the kimberlites?

The diamonds are predominantly inclusion-bearing xenocrysts in the kimberlites that are made available by industry for study – info added to first section “Database” in Methods

-Line 403: specify that Sc is homovalent and trivalent, i.e. analogous only to trivalent V
Sentence has been removed to tighten the text

-Line 410: ppm or wt.%? I’m confused
ppm - corrected

-Line 430: “The” two data-sets “used in this study”
Done

-Line 445: for both “types of sample”
Done

-Line 468: within “uncertainties”
Done

Unsolicited edits:

1. Finding a systematic deviation of V abundances in minerals analysed by both EPMA and LAM-ICPMS in two data-sets generated at University of Alberta, we initially suggested that this may reflect an overestimate of EPMA-derived V abundances due to oxidation of the V metal standard (Methods). Dr. A. Locock has now revisited this issue and analysed magnetite by EPMA using either V metal or YVO₄ as the primary V standard. He finds that the results agree within uncertainty, without a systematic bias, suggesting that the standards are good. Following up further on this issue, Stachel finds that V analyses using the old quadrupole instrument at University of Alberta are underestimated, due to an overcorrection of oxide interferences on transition metals applied by the operator of that instrument at the time. The Methods section has been changed to reflect these new insights. This has prompted us to further investigate and document accuracy of LAM-derived data. Reference materials were explicitly stated to have been measured as unknowns in all but two studies, although values are not always reported. When measured values are given, they agree with accepted values within uncertainty. We have added this information to Table S1 and now use EPMA rather than LAM data for Smit et al. 2014 (EPMA data were already used for Stachel et al. 2018 as

only 2/8 eclogite DI had both EPMA and LAM data, the remainder used in the comparison being for peridotitic DI). This change propagates into some statistical values given in Tables 1 and 2 and average uncertainties in the Methods, but does not affect the conclusions reached in the paper.

2. We added 5 DI data points from Donnelly et al. 2007 for completeness (none paired, i.e. no additional temperatures, reconstructed bulk rocks or distribution coefficients = no changes to figures in main text). This led to additions to some supplementary figures and very small changes to DI statistics, but had no effect on the conclusions reached in the paper.
3. We made minor edits throughout the revised main text and SI to avoid repetition and further improve accuracy, readability and clarity.

REVIEWERS' COMMENTS

Reviewer #1 (Remarks to the Author):

The authors' have thoroughly addressed all of reviewers' comments in their re-submitted manuscript, which is now suitable for publication in Nature Communications.

Reviewer #3 (Remarks to the Author):

You have improved the manuscript as I requested, and I am satisfied with it now. I would now recommend publication.

Point-by-point response to the reviewers' comments

Reviewer comments are reproduced verbatim below in black font, our replies in blue font.

Reviewer #1 (Remarks to the Author):

The authors' have thoroughly addressed all of reviewers' comments in their re-submitted manuscript, which is now suitable for publication in Nature Communications.

We are happy we satisfactorily addressed the reviewers' comments, and thank them for their time.

Reviewer #2 (Remarks to the Author): not provided

Reviewer #3 (Remarks to the Author):

You have improved the manuscript as I requested, and I am satisfied with it now. I would now recommend publication.

We appreciate the reviewer's recommendation and time they spent on helping improve the work.